# The Influence of Zinc Oxide Nanoparticles and Salt Stress on the Morphological and Some Biochemical Characteristics of *Solanum lycopersicum* L. Plants

**DOI:** 10.3390/plants13101418

**Published:** 2024-05-20

**Authors:** Mostafa Ahmed, Diaa Attia Marrez, Roquia Rizk, Mostafa Zedan, Donia Abdul-Hamid, Kincső Decsi, Gergő Péter Kovács, Zoltán Tóth

**Affiliations:** 1Festetics Doctoral School, Institute of Agronomy, Georgikon Campus, Hungarian University of Agriculture and Life Sciences, 8360 Keszthely, Hungary; mostafa.ahmed.abdelmagid@agr.cu.edu.eg; 2Department of Agricultural Biochemistry, Faculty of Agriculture, Cairo University, Giza 12613, Egypt; roquiaibrahim@gmail.com; 3Food Toxicology and Contaminants Department, National Research Centre, Dokki, Cairo 12622, Egypt; diaamm80@hotmail.com; 4Institute of Agronomy, Georgikon Campus, Hungarian University of Agriculture and Life Sciences, 8360 Keszthely, Hungary; toth.zoltan@uni-mate.hu; 5National Institute of Laser Enhanced Science, Cairo University, Giza 12613, Egypt; mostafa.zedan@niles.edu.eg; 6Heavy Metals Department, Central Laboratory for The Analysis of Pesticides and Heavy Metals in Food (QCAP), Dokki, Cairo 12311, Egypt; donia.atalah11@gmail.com; 7Institute of Agronomy, Szent István Campus, Hungarian University of Agriculture and Life Sciences, 2100 Gödöllő, Hungary; kovacs.gergo.peter@uni-mate.hu

**Keywords:** zinc oxide nanoparticles, tomato, salinity, abiotic stress, phenolic compounds, biochemical parameters

## Abstract

Salinity reduces crop yields and quality, causing global economic losses. Zinc oxide nanoparticles (ZnO-NPs) improve plant physiological and metabolic processes and abiotic stress resistance. This study examined the effects of foliar ZnO-NPs at 75 and 150 mg/L on tomato Kecskeméti 549 plants to alleviate salt stress caused by 150 mM NaCl. The precipitation procedure produced ZnO-NPs that were characterized using UV-VIS, TEM, STEM, DLS, EDAX, Zeta potential, and FTIR. The study assessed TPCs, TFCs, total hydrolyzable sugars, total free amino acids, protein, proline, H_2_O_2_, and MDA along with plant height, stem width, leaf area, and SPAD values. The polyphenolic burden was also measured by HPLC. With salt stress, plant growth and chlorophyll content decreased significantly. The growth and development of tomato plants changed by applying the ZnO-NPs. Dosages of ZnO-NPs had a significant effect across treatments. ZnO-NPs also increased chlorophyll, reduced stress markers, and released phenolic chemicals and proteins in the leaves of tomatoes. ZnO-NPs reduce salt stress by promoting the uptake of minerals. ZnO-NPs had beneficial effects on tomato plants when subjected to salt stress, making them an alternate technique to boost resilience in saline soils or low-quality irrigation water. This study examined how foliar application of chemically synthesized ZnO-NPs to the leaves affected biochemistry, morphology, and phenolic compound synthesis with and without NaCl.

## 1. Introduction

On a global scale, there is significant consumption of both fresh and processed tomatoes (*Solanum lycopersicum* L.). Therefore, a widely consumed functional food can be ingested by individuals worldwide, either in its uncooked or cooked state, or as a primary ingredient that undergoes processing to create various products such as tomato powder, juice, puree, sauces, ketchup, and paste [1,2]. These tomato fruits contain a higher concentration of bioactive compounds and nutrients than other fruits [3,4] that are good for the skin and body and may help treat or prevent several chronic degenerative disorders in people [3].

Every year, a multitude of factors can impact the nutritional quality and quantity of tomatoes [5]. Agricultural crop productivity and cultivation are impacted by a variety of environmental difficulties, including strong winds, extreme temperatures, soil salinity, drought, and flooding [6]. One of the harshest ecological stresses on crop quality and yield is salinity in the soil [7,8]. The onset of the 21st century was marked by environmental pollution, a worldwide scarcity of water resources, and an increase in the salinization of soil and water. The sustainability of agriculture is at risk due to the increasing population and the decreasing availability of arable land [8]. The physiological and biochemical pathways of crop plants are negatively impacted by soil salt through intricate mechanisms [9]. Tomatoes have a moderate sensitivity to soil salinity compared to other vegetable crops. The greatest threshold of soil sodicity for yield loss in tomatoes is 2.5 dS m^−1^. The abundance of diverse germplasm, including a wide range of wild species, is valuable for incorporating characteristics that confer resistance to various diseases, as well as tolerance to soil salinity and drought [10]. The tomato variety employed in the study, Kecskeméti 549, is a medieval cultivar known for its determinate growth. Tomato berries mature uniformly in a single hue, possess a somewhat elongated and cylindrical shape, and may be effortlessly detached from the stem by pinching. The typical weight of their berries ranges from 50 to 60 g [11]. Salt stress can cause a variety of problems at multiple levels, including molecular, morphological, and biochemical (metabolic) disruptions. Several regulatory elements are responsible for reducing the negative consequences of abiotic stress in plants [12].

Researchers from several sectors, such as agriculture and health, have recently shown significant interest in green, biogenic, or chemically produced nanoparticles. These nanoparticles are valued for their practicality, low toxicity, long-lasting effects, and affordable price. Nanoparticles are extensively utilized to protect and enhance plant growth and development by mitigating various abiotic stresses such as salt, freezing, flooding, drought, heavy metal toxicity, and excessive heat [13]. It has been observed that zinc oxide nanoparticles (ZnO-NPs) have important and essential functions in promoting plant growth and enhancing plant resistance to salt stress in several plant species [14]. Zinc (Zn^2+^) is a crucial micronutrient that is necessary for the proper functioning of living organisms’ metabolism [15]. It carries out vital duties by participating in the activities of many enzymes [16] and serves as a regulatory cofactor in protein synthesis. Moreover, a lack of Zn^2+^ results in the decrease of multiple metabolic processes, including growth, ultimately affecting crop yields [17].

The application of ZnO-NPs is widely utilized in several fields, including agriculture, solar cells, lotions, paints, rubber, concrete, cosmetics, and medicine [18]. A study has shown that the use of ZnO-NPs improves growth by controlling photosynthetic processes and the production of active oxygen species such as superoxide and hydroxide anion [19]. Furthermore, a separate study showed that ZnO-NPs increased the concentration of chlorophyll, carotenoids, protein, and the activity of antioxidative enzymes in cotton plants [20]. Additionally, metal/metal oxide nanoparticles have diverse applications, notably in the food business [21]. Applications in the food industry may involve the creation of novel biodegradable packaging materials that possess antibacterial qualities, as well as the development of protective coatings for work surfaces, specifically in the context of meat processing [22,23,24]. A liquid-phase nanocomposite comprising ZnO nanoparticles (NPs) and fluoroplastic was created, which is suitable for application on several surfaces, including sputtering. When the mixture hardens, it creates a strong wall that stops the growth of important medical Gram-positive bacteria (*Listeria monocytogenes* and *Staphylococcus aureus*) and Gram-negative bacteria (*Pseudomonas aeruginosa* and *Staphylococcus typhimurium*) [22,25]. Furthermore, cultures of eukaryotic cells do not experience any harmful effects from the produced composite [22,26]. The food industry has used a similar nanomaterial to treat cutting boards in meat processing facilities [25,26]. The objective of the present study was to investigate the effects of applying chemically synthesized ZnO-NPs to the leaves on the biochemistry, morphology, and formation of phenolic compounds, both in the presence and absence of NaCl.

## 2. Results

### 2.1. Characterization of Chemically Synthesized ZnO-NPs

Figure 1 displays the spectrum of a distinct absorption peak of ZnO at a wavelength of 370 nm, which corresponds to the intrinsic band-gap absorption of ZnO. This absorption is caused by electron transitions from the valence band to the conduction band (O_2p_→Zn_3d_).

Figure 2, Figure 3 and Figure 4 and Table 1 display the transmission electron microscopy (TEM), scanning electron microscopy (SEM), energy-dispersive X-ray spectroscopy, and size distribution of the zinc oxide nanoparticles (ZnO-NPs) that were synthesized using the precipitation method. The transmission electron microscopy (TEM) image (Figure 2) revealed that the ZnO nanoparticles (NPs) have developed in a nearly hexagonal morphology, indicating the high quality of the ZnO-NPs. Figure 3 displays the scanning electron microscope (SEM) images of the ZnO-NPs, revealing a uniform shape and size for the particles. Furthermore, the presence of ZnO-NPs in the powder form is evident. Figure 4 confirms the primary constituents of the ZnO sample consist of zinc (Zn) and oxygen (O), which are evenly distributed across the surface of the ZnO nanoparticles. The size distribution of the ZnO-NPs produced using the precipitation method at a calcination temperature of 200 °C for 2 h is monomodal with a half-width of approximately 41.166 nm.

Figure 5 displayes the FTIR spectrum of chemically produced zinc oxide nanoparticles, obtained within the wavelength range from 3900 to 300 cm^−1^. The spectrum of zinc oxide nanoparticles exhibited peaks corresponding to six functional groups at 3150, 1631.48, 1425.14, 1078.98, 447.404, and 370.23 cm^−1^.

Figure 6 displays the X-ray diffraction (XRD) pattern of chemically generated zinc oxide (ZnO) nanoparticles, both chemically and green. The diffraction peaks were seen at the following angles (2θ): 31.73°, 34.38°, 36.23°, 47.58°, 56.58°, 62.88°, and 76.92°.

Zeta potential analysis was conducted to identify the surface charges accumulated by zinc oxide nanoparticles, providing additional information on the stability of the resulting colloidal zinc oxide nanoparticles. The magnitude of Zeta potential indicates the possible stability of colloids. The nanoparticles produced in that study were highly stable, as evidenced by a Zeta potential value of −30.214 mV. The Zeta potential of the chemically produced zinc oxide nanoparticles was measured in water, using it as a dispersant. The measured Zeta potential was determined to be −30.214 mV, as illustrated in Figure 7.

### 2.2. Assessment of the Elements, Growth, and Chlorophyll Attributes in Different Treatments from Tomato

In the current study, three macro and three other microelements were determined as shown in Table 2. Under severe salt stress (150 mM NaCl), the highest sodium (Na) contents were measured from the fourth stressed treatment (T4) in a concentration of 26.48 ± 0.21, followed by the T5 and T6 treatments in concentrations of 13.51 ± 0.06 and 13.27 ± 0.23 µg/g, respectively, without any noticeable significant difference between the ZnO-NPs foliar sprayed treatments (T5 and T6). The highest potassium (K) contents were obtained from T1–T3 treatments, and they varied between 29.20 ± 0.22 and 29.02 ± 0.02 µg/g. The highest zinc (Zn) contents were obtained from the ZnO-NPs foliar sprayed T2, T3, T5, and T6 treatments, and the copper (Cu) content was noticed to be a little bit higher in the stressed treatments than the non-stressed treatments. In addition, the magnesium (Mg) and manganese (Mn) contents were obtained from the six different treatments without any significant differences.

The chlorophyll content (soil plant analysis development; SPAD units) in the different treatments (Figure A1 and Figure A2) was determined as shown in Table 3. It was determined in four different stages to examine how it would be influenced by the NaCl stress and foliar spraying of chemically synthesized zinc oxide nanoparticles. It was observed that the readings of the SPAD measuring the chlorophyll content in the different treatments were not significantly different in the first two stages, but the chlorophyll content decreased in a time manner for the stressed treatments in the third stage even if they were sprayed by ZnO-NPs. On the other side, the chlorophyll content increased in the non-stressed treatments (T1–T3). It was noticed that the non-stressed treatments (T1–T3) had higher chlorophyll content than the stressed treatments (T4–T6) in the third stage, but the content of chlorophyll began to rise again in the fifth and sixth stressed treatments that were sprayed with ZnO-NPs in the fourth stage. However, in the fourth treatment that was not sprayed with ZnO-NPs, the chlorophyll content did not significantly rise. It was clear that there was a manner of increasing the chlorophyll content in the stressed treatments when they were sprayed with ZnO-NPs.

The tomato plant height (cm) in the different treatments was determined as shown in Table 4. Like the chlorophyll content, it was determined in four different stages as an indication of how growth and morphological attributes would be influenced along the experiment. It was observed that the tomato plant heights in the different treatments were not significantly different in the first two stages. The tomato plants showed an increase in height according to the six different treatments, but it was observed that the rate of increase in the non-stressed treatments (T1–T3) was higher than in the stressed treatments (T4–T6). In the third stage, the increasing rate of tomato plants’ height for the fifth and sixth stressed treatments (T5 and T6) was close to the rate of elongation in the non-stressed treatments (T1–T3), as they were sprayed by ZnO-NPs. On the other side, the increasing rate of tomato plants’ height for the fourth stressed treatment (T4) was lower than in the other stressed treatments (T5 and T6).

The tomato stem width (cm) in the different treatments was determined as shown in Table 5. Like the previous growth and morphological attribute (plant height), stem width was determined in four different stages. It was observed that the tomato plants’ stem width in the different treatments was not significantly different in the first stage. The tomato plants showed an increase in stem width according to the six different treatments. By the second stage, it was observed that the rate of increase in the non-stressed treatments (T1–T3) was higher than in the stressed treatments (T4–T6). In the third stage, the increasing rate of tomato stem width for the stressed treatments (T5-T6) was lower than the rate of elongation in the non-stressed treatments (T1–T3), but the fifth and sixth treatments (T5 and T6) were sprayed with different concentrations of ZnO-NPs three times, so they showed a little bit of increase compared to the fourth treatment (T4).

The tomato leaf area (cm^2^) in the different treatments was determined as shown in Table 6. The leaf area was determined in four different stages as an indication of the influence of NaCl stress and foliar spraying of chemically synthesized zinc oxide nanoparticles. It was observed that the readings in the different treatments were not significantly different. There was an increasing manner in the leaf area within the different stages, and there was an obvious increase in the leaf area of the non-stressed treatments (T1–T3) during the study compared to the stressed treatments (T4–T6).

### 2.3. Phenolic Profile

The HPLC analysis of tomato leaves’ extracts for determining their phenolic compounds was determined as shown in Figure A3, Figure A4, Figure A5, Figure A6, Figure A7, Figure A8 and Figure A9 and Table 7. Figure A3 shows the chromatogram of the prepared standards that expressed 20 compounds. Three compounds out of those twenty were not found in any of the examined tomato extracts from the different treatments. These three compounds were genetisic acid, rosmarinic acid, and kampferol. It was noticed that the obtained phenolic compounds from the stressed treatments (T4–T6) were higher in concentration with totals of 937.62, 1223.78, and 1309.78 µg/g in T4–T6 treatments, respectively. The concentration of the compounds obtained from the non-stressed treatments (T1–T3) had totals of 562.02, 753.02, and 937.62 µg/g in T1–T3 respectively. Figure A4, Figure A5, Figure A6, Figure A7, Figure A8 and Figure A9 shows the presence of gallic acid, protocatechuic acid, *p*-hydroxybenzoic acid, catechin, chlorogenic acid, caffeic acid, syringic acid, vanillic acid, ferulic acid, sinapic acid, rutin, *p*-coumaric acid, apigenin-7-glucoside, cinnamic acid, quercetin, apigenin, and chrysin.

### 2.4. Determination of Different Biochemical and Stress Markers in the Leaves from Different Treatments of Tomato

Table 8 shows the results: as the salt concentrations increased in the tomato leaves, the total phenolics and flavonoids (TPCs and TFCs), total hydrolyzable sugars, total free amino acids, proline, hydrogen peroxide (H_2_O_2_), the malondialdehyde (MDA) contents also increased. The effects of treatments on plant contents of the previous stress and biochemical markers were found to be statistically significant. The fourth stressed treatment (T4) had the highest concentrations of the aforementioned parameters compared to the two other stressed treatments (T5 and T6) except with the protein content. Those treatments (T5 and T6) were sprayed with 75 and 150 mg/L zinc oxide nanoparticles solutions. So, the secretion of the above-mentioned compounds was controlled to be in a compatible manner with the non-stressed treatments (T1–T3). The fourth treatment (stressed) (T4) had the highest values of the estimated parameters among the other stressed or non-stressed treatments (T1–T3, T5, and T6), and it showed concentrations of 4043 ± 0.44, 1191.83 ± 16.66, 155.97 ± 0.90, 986.68 ± 8.61, 71.54 ± 2.60, 1158.76 ± 11.40 µg/g, and 11.77 ± 0.24 mmols/mL for TPCs, TFCs, total hydrolyzable sugars, total free amino acids, proline, H_2_O_2_, and MDA contents, respectively. However, for protein content, the first treatment (T1) had the highest value in a concentration of 81.28 ± 1.16 µg/g. The lowest concentrations of the previously estimated biochemical and stress markers were 2096 ± 0.10 µg/g of TPCs, 401 ± 3.39 µg/g of TFCs, 84.58 ± 4.10 µg/g of total hydrolyzable sugars estimated as glucose, 163.73 ± 2.92 µg/g of total free amino acids estimated as L-leucine, 15.23 ± 0.64 µg/g of proline, 418.76 ± 1.78 µg/g of H_2_O_2_, and 1.56 ± 0.07 mmols/mL of MDA in the first non-stressed treatment (T1), but 26.11 ± 0.33 µg/g of protein estimated as BSA in the fourth treatment (T4).

## 3. Discussion

### 3.1. Characterization of Chemically Synthesized ZnO-NPs

The UV-visible absorption spectrum was primarily employed to analyze the optical characteristics of nanoparticles [27]. This optical property aligns with the optical activity of ZnO-NPs that were synthesized using a solvothermal technique [28]. Furthermore, this distinct peak indicated that the particles were of nanoscale dimensions, and the distribution of particle sizes was highly concentrated. The shape, size, and elemental distribution of the obtained chemically synthesized ZnO-NPs were characterized using the STEM, TEM, EDAX, and DLS techniques. The results from Figure 2, Figure 3 and Figure 4 and Table 1 confirm that ZnO-NPs generated using the precipitation process employing sodium hydroxide and zinc nitrate hexahydrate have a nearly hexagonal morphology, and limited size distribution of 41.166 nm, distinguishing them from other results previously documented in other research, compared to some of the other results [28,29,30]. To obtain additional insight into the topographies of ZnO-NPs, the EDAX analysis of the sample was performed from the same area as shown in Figure 3. The EDAX examination confirmed the presence of zinc and oxygen in a percentage of 67.55 and 32.02%. The detection of copper in the EDAX analysis is attributed to the use of the lacy carbon-coated copper grid, which enhances the quality of the SEM pictures [31].

Using Fourier transform infrared spectroscopy, the sample showed six functional groups detected at 3150, 1631.48, 1425.14, 1078.98, 447.404, and 370.23 cm^−1^ (Figure 5). The intense and wide peak observed at 3150 cm^−1^ in the upper region was a result of the stretching vibration of hydroxyl (OH) groups [32,33]. The peak observed at 1631.48 cm^−1^ was a result of the presence of the C=O amide I and amide II functional groups. The bending vibration band of the C-H bond occurred at a frequency of 1425.14 cm^−1^. The C–O stretching vibration band occurred at a frequency of 1078.98 cm^−1^. The frequencies of 447.404 and 370.23 cm^−1^ corresponded to the stretching vibration of Zn–O bonds, providing evidence for the synthesis of the product. FTIR analysis was conducted to determine the potential biomolecules accountable for the bioreduction of zinc oxide nanoparticles. The prior results are consistent with the findings reported in other papers [28,29,30,31,34].

The XRD is a rapid test providing information on crystal size and structure [35]. In Figure 6, the XRD analysis confirmed that the diffraction peak planes of the precipitated ZnO-NPs were matched well with the quartzite ZnO reported in the joint committee on powder diffraction standards (JCPDS) data [36]. They represent the reflection resulting from the crystal planes of the hexagonal wurtzite structure of ZnO nanoparticles. Zinc oxide exists in two primary crystalline structures: hexagonal wurtzite and cubic zincblende. The wurtzite structure is the most stable and prevalent under normal conditions. The presence of strong peaks suggests that the ZnO nanoparticles exhibited a high degree of crystallinity. The XRD patterns did not exhibit diffraction peaks associated with the contaminant, thereby validating the excellent purity of the produced nanoparticles. The observed results are consistent with the findings of other authors who have examined the produced zinc oxide nanoparticles using the XRD diffractometer [34,37].

When the particles in a suspension possess significant negative or positive Zeta potential values (Figure 7), they will exhibit repulsive forces toward one other, preventing the aggregation of nanoparticles. Conversely, when particles possess low Zeta potential levels, no opposing force hinders their aggregation and subsequent coming together. Zeta potential values exceeding +30 mV or falling below −30 mV are typically associated with the formation of stable suspensions [38]. The Zeta potential of the chemically produced zinc oxide nanoparticles was measured in water using it as a dispersant. The measured Zeta potential was determined to be −30.214 mV, as illustrated in Figure 7. The Zeta potential is directly related to the stability of the nanoparticles while they are in a solution. The nanoparticles produced in this work were highly stable, as evidenced by a Zeta potential value of −30.214 mV [39].

### 3.2. Assessment of the Elements, Growth, and Chlorophyll Attributes in Different Treatments from Tomato

Salinity results in significant economic losses in agricultural output due to crop yield and quality reductions. The prevailing consensus is that a significant portion of cultivable agricultural land will become inoperable in the future as a result of salt. Hence, this study aimed to examine the impact of zinc oxide nanoparticle influence on tomato plants in the presence of salt conditions, and the results obtained are given in Table 2, Table 3, Table 4, Table 5, Table 6 and Table 7 and Figure A1 and Figure A2. It is possible for plant pores to more easily access and absorb nanosized nutrients, which ultimately results in higher efficiency [40]. Zn^2+^ is a cofactor for several enzymes, and the size of its particles makes it possible for it to be soluble and to enter the leaf surface, where it can then release Zn^2+^ ions via the cuticle [41]. Zinc oxide nanoparticles (ZnO-NPs) have been found to enhance nutritional absorption, regulate the ratio of sodium to potassium ions (Na^+^/K^+^), maintain water balance, enable ion accumulation, and mitigate the negative impacts of abiotic stressors like salinity stress [7].

K^+^ and Na^+^ ions act as antagonists: when a plant absorbs a large amount of Na^+^, it displaces K^+^ ions from the available Donnan binding sites, resulting in a relative deficit of K^+^ in the plant [42]. Insufficient potassium results in the withering of plants, loss of moisture, and impaired water equilibrium [43]. Plants exhibit stunted growth and display dead white or brown patches as a result of cellular shrinkage and tissue collapse, and potassium is the element that is needed in the second highest amount, as plants take it in the form of K^+^ ions [44]. Potassium stimulates about 40 enzymes, influences hydration, and has a crucial function in signal transmission, and adequate provision of potassium is essential for numerous metabolic and physiological processes [45]. K^+^ enhances the production of complex carbohydrates and improves crop resilience, hence mitigating crop damage caused by factors such as wind, water scarcity, cold, and pathogens. To protect against wind damage, the plant activates a process called enhanced high-molecule carbohydrate biosynthesis, which leads to the build-up of cellulose that increases the fiber content of the plant’s tissues, making them stronger [46]. Potassium enhances protein synthesis by facilitating the interaction between mRNAs and the ribosome [47]. It exerts a beneficial influence on the process of photosynthesis and the movement of materials within the plant, enhances the water utilization efficiency of plants, and plays a crucial part in the process of plant evaporation and respiration as it decreases the magnitude of evaporation [48].

The current study showed a significant decrease in potassium content in the fourth stressed treatment that was not sprayed with ZnO-NPs. However, the exogenous application of ZnO-NPs improved the element concentration in tomato plants of the stressed treatments that were sprayed with ZnO-NPs (T5 and T6) (Table 2). ZnO-NPs were effective in the nutrient balance in plants and the survival of the plant under NaCl stress, as the contents of the mineral Na were decreased in the foliar sprayed stressed treatments (T5 and T6) compared to the non-sprayed stressed treatment (T4) [49]. Copper is predominantly concentrated in chloroplasts, accounting for 70% of its presence. It is an integral part of plastocyanins, which are proteins that contain copper. It has a function in the electron transport chain during photosynthesis. The individual was responsible for the development of a highly significant antioxidant enzyme, known as superoxide dismutase, in conjunction with the element zinc [50]. The Cu^2+^ content showed a significant difference between the stressed treatments and the non-stressed treatments [51]. On the other side, there was not any significant difference between the stressed and non-stressed treatments in the concentrations of the two elements Mg and Mn, even if they were sprayed with ZnO-NPs or not, and these results were reinforced by a previous research article on *Ginkgo biloba* [52]. The previous findings could be attributed to the impact of ZnO-NPs on minerals in plants, which is contingent upon the concentration. It was confirmed that a noteworthy rise in the levels of zinc ions (Zn^2+^) in the leaves and seeds when 40 ppm of ZnO-NPs was applied to the foliage [53]. In a recent study, it has been shown that ZnO-NPs stimulated the uptake of K^+^, Zn^2+^, and Cu^2+^ in faba bean [54] and rapeseed [55] under salinity stress and substituting the Na^+^ with the previous minerals, thus decreasing the phytotoxicity by Na^+^. 

In the current study, the effect of severe salt stress (150 mM NaCl) on the chlorophyll content, plant height, stem width, and leaf area was detrimental to tomato plants depending on the different concentrations used to spray the plants with ZnO-NPs (75 and 150 mg/L). A decrease was observed in the chlorophyll content, plant height, stem width, and leaf area at the severe concentration of NaCl (150 mM) [56]; however, for this inhibitory effect of salt stress on chlorophyll content, and the growth attributes—plant height, stem width, and leaf area in tomato plants—it was noticed that harmful effect was less when ZnO-NPs foliar spray was applied.

SPAD value is an indicator of chlorophyll content, which displays the functions of photosynthetic apparatus [57]. The current study confirmed that the chlorophyll content exhibited a considerable drop in correlation with the severe salt concentration, with the most pronounced reduction found in plants cultivated under 150 mM NaCl (Table 3). Shin et al. found that the levels of chlorophyll a and chlorophyll b in tomatoes reduced in proportion to high salt concentrations [58]. Additionally, salt stress adversely affects the SPAD value in cucumber plants [59]. In the present study, it was found that the content of chlorophyll began to rise again in the fifth and sixth stressed treatments that were sprayed with ZnO-NPs (75 and 150 mg/L) in the fourth stage of collecting the readings, but in the fourth stressed treatment that was not sprayed with ZnO-NPs (T4), the chlorophyll content did not significantly rise. It was clear that there was a manner of increasing the chlorophyll content in the stressed treatments when they were sprayed with ZnO-NPs. In a study by Mahawar et al. (2024), it was primarily observed the combined impact of NaCl and zinc oxide nanoparticles on the photochemistry of the photosystems (PSII and PSI) at a higher salt concentration of 300 mM [57]. Generally, applying metal oxide nanoparticles to plant leaves and subjecting them to higher NaCl treatment enhanced the plant’s ability to convert sunlight into energy by speeding up the flow of electrons from active reaction centers to the quinone pool. This resulted in an increase in the efficiency of photosystem II, a decrease in the amount of light that is not used for photosynthesis, and a reduction in the stress placed on the cytochrome complex [57]. Foliar zinc oxide nanoparticles (ZnO-NPs) have been observed to be more efficient in improving the effectiveness of photosystem II (PSII) photochemistry. Zinc is a crucial element that is recognized for its ability to enhance photosynthetic activities in plants under stress [60]. The replacement of zinc ions in the soil is not possible, making it crucial to restore them through foliage [61]. The zinc ion, an essential nutrient, greatly influences the synthesis of auxin, metabolic processes related to nitrogen, and the proliferation of root cell tissue. This is because zinc enhances the biosynthesis of tryptophan [7,62]. Indole-based auxins are produced from tryptophan; therefore, the zinc nanoparticles will have an indirect impact on the elongation growth [63]. This is because it functions as a cofactor for a range of enzymes, including oxidases, dehydrogenases, and anhydrases, such as peroxidases. Zinc enhances the cation-exchange capacity of the root, facilitating the absorption of vital nutrients, such as nitrogen. Consequently, this leads to an increase in the overall quantity of protein. In addition, zinc plays a crucial role in the metabolism by facilitating the breakdown of carbohydrates and proteins [7,64]. The presence of zinc oxide nanoparticles can modify the photosynthetic systems when plants are exposed to NaCl stress. This modification occurs by stimulating the enzymes responsible for the photosynthetic electron transport and water-splitting processes [65].

The foliar spray of ZnO-NPs had a substantial impact on the plant height, stem width, and leaf area of the tomato Kecskeméti 549 variety, as indicated by the data presented in Table 4, Table 5 and Table 6. That positive impact of the ZnO-NPs was thought to be because of increasing the tryptophan amino acid that increased the production of the indole-based auxins [63]. The plant reached its maximum height of 75.25 ± 2.10 cm at the fourth stage of the experiment, namely in the third treatment (T3), where a concentration of 150 mg/L of ZnO-NPs was used (Table 4). This height was greater than that seen in the control group. The fourth treatment (T4), with a concentration of 150 mM NaCl, resulted in the lowest plant height, measuring 57.00 ± 2.42 cm. The utilization of zinc oxide nanoparticles may enhance plant growth, particularly in terms of plant height, by facilitating the release of essential nutrients for crop development [66]. In their study, Sun et al. (2020) found that applying a solution of 100 mg/L ZnO-NPs through foliar spraying resulted in the highest number of tomato plants [67]. The tomato plants that were treated with a concentration of 50 parts per million (ppm) of zinc oxide nanoparticles (ZnO-NPs) showed the greatest increase in shoot length, with a growth increment of 30.1% compared to the control group [68].

The plant reached its maximum stem width of 1.20 ± 0.02 cm at the fourth stage of the experiment, namely in the second treatment (T2), where a concentration of 75 mg/L of ZnO-NPs was applied (Table 5). This width was a little greater than that seen in the control group. The fourth treatment (T4), with a concentration of 150 mM NaCl, resulted in the lowest plant stem width, measuring 0.78 ± 0.03 cm. In the case of leaf area (Table 6), the increased leaf area (638.10 ± 5.140 cm^2^) was obtained from the plant receiving 75 mg/L ZnO nanoparticles (T2) followed by the plant receiving 150 mg/L ZnO-NPs (T3) with 632.15 ± 17.15. The lowest result (452.75 ± 40.25 cm^2^) was in the fourth treatment (T4), which was severely stressed with 150 mM NaCl and did not receive any foliar spray with zinc oxide nanoparticles. ZnO-NPs are highly efficient in providing Zn^2+^ since they play a crucial role in supporting several aspects of plant growth and development, such as glucose and protein metabolism. Zinc ions participate in the formation of enzyme-substrate complexes, and bind NAD to the surface of the apoenzyme in the plants [69]. Additionally, they aid in auxin synthesis, which eventually promotes the development of plant cell walls and cell differentiation [70].

The decreased reduction of tomato stem under salinity can be related to the excessive accumulation of Na^+^ and Cl^−^ ions in many cellular compartments of both the root and aerial plant organs [71,72]. The accumulation of these ions to harmful levels disrupts the process of genetic expression, protein synthesis, enzyme activity, energy consumption, and cell division. It also harms the cellular ultrastructure and can finally lead to cell death [7,73]. Salinity not only leads to ionic toxicity but also disrupts osmotic balance, denatures cell membranes, causes leakage of osmolytes, and disrupts nutritional balance. As a result, it affects turgor pressure, cell elongation, and the morphological characteristics of plants under stress [72,74]. While many plants can regulate their osmotic balance by producing suitable organic electrolytes through biosynthesis, this process can require up to 10 times more energy. This energy expenditure is further exacerbated when plants are continuously exposed to hypertonic solutions [75]. Seleiman et al. (2023) demonstrated that applying ZnO-NPs to the leaves, particularly at a concentration of 100 mg/L, effectively improved the negative effects of salt on growth metrics [75]. Zinc, supplied by ZnO-NPs, stimulates the production of endogenous plant regulators and growth promoters including indole-3-acetic acid (IAA) and gibberellic acid (GA3). These substances have a role in metabolic activity, cell elongation, and cell division, ultimately leading to improved plant growth [76,77].

### 3.3. Phenolic Profile (HPLC, TPCs, and TFCs)

Cellular respiration is accelerated due to the impact of stress, causing a shift in biochemical pathways from glycolysis to the pentose phosphate pathway. The secondary metabolic pathways, such as the Shikimic acid pathway and phenylpropanoid biosynthesis pathway, that originate from this point, experience an increase in speed. This acceleration leads to the production of secondary metabolites, such as phenolic compounds and antioxidants, which aid in the cellular-level regeneration processes of plants [78,79]. Figure A3, Figure A4, Figure A5, Figure A6, Figure A7, Figure A8 and Figure A9 and Table 7 showed the HPLC analysis of tomato leaves’ extracts for quantifying their phenolic compounds. The tomato leaves from different treatments (stressed and non-stressed) reached their maximum secretion of the phenolic compounds through chlorogenic acid of the levels 154.08, 239.25, 355.54, 359.51, 617.96, and 603.38 µg/g, followed by gallic acid of the following levels: 122.91, 149.63, 141.73, 206.62, 192.26, and 203.61 µg/g, and these results for chlorogenic and gallic acids were for T1, T2, T3, T4, T5, and T6, respectively. It was noticed that between the stressed treatments, the fourth stressed treatment (T4) had the highest content of gallic acid, and the two other stressed treatments (T5 and T6) that were sprayed by ZnO-NPs in concentrations of 75 and 150 mg/L had lower concentrations than the fourth treatment. However, for chlorogenic acid, it was noticed that between the stressed treatments, the fourth stressed treatment (T4) had the lowest content of chlorogenic acid, and the two other stressed treatments (T5 and T6) that were sprayed by ZnO-NPs in concentrations of 75 and 150 mg/L had higher concentrations than the fourth treatment. On the other side, the tomato leaves from different treatments (stressed and non-stressed) reached their minimum secretion of the phenolic compounds through apigenin of the following levels: 0.19, 1.19, 5.50, 4.62, 6.79, and 5.20 µg/g, and these results were for T1, T2, T3, T4, T5, and T6, respectively. It was noticed that between the stressed treatments, the fourth stressed treatment (T4) had the lowest content of apigenin, and the two other stressed treatments (T5 and T6) that were sprayed by ZnO-NPs in concentrations of 75 and 150 mg/L had higher concentrations than the fourth treatment. So, in the current study, it was confirmed that the NaCl stress had a positive effect on secreting the phenolic compounds, and the trend of the highest values of the produced compounds was toward the stressed treatments (T4–T6) even if they were sprayed with ZnO-NPs with their different concentrations or not.

Table 8 also shows the total phenolic and flavonoid contents. The tomato leaves from different treatments (stressed and non-stressed) reached their maximum secretion of the TPCs at the level of 4043 ± 0.44 µg/g in the fourth stressed treatment (T4), followed by the following levels: 3730 ± 0.32 and 2596 ± 0.11 µg/g in the fifth and sixth treatments, respectively. The tomato leaves reached their minimum production of TPCs in the first non-stressed treatments at the level of 2096 ± 0.10 µg/g. The previous results matched the results from HPLC detecting gallic acid in the different tomato treatments. It was also noticed that ZnO-NPs were highly efficient in regulating the secretion of the phenolic compounds in the stressed treatments to grow closer to the levels of the non-stressed treatments; this was more obvious in the fifth and sixth treatments compared to the fourth one. On the other side, the tomato leaves from different treatments (stressed and non-stressed) reached their maximum secretion of the TFCs at the level of 1271.37 ± 79.35 µg/g in the sixth stressed treatment (T6), followed by the following levels: 1222.39 ± 62.84 and 1191.83 ± 16.66 µg/g in the fifth and fourth treatments, respectively. The tomato leaves reached their minimum production of TPCs in the second non-stressed treatment at the level of 356.56 ± 37.23 µg/g. The previous showed a different manner than it was with the TPCs. It was observed that the secretion of the total flavonoids increased in the stressed treatments that were sprayed with different concentrations of ZnO-NPs compared to the content in the fourth stressed treatment (T4) that was not sprayed with ZnO-NPs. In the current study, it is thought that ZnO-NPs were efficient in increasing the secretion of the flavonoid content in the stressed treatments (T5 and T6).

Phenolic compounds play a crucial role in neutralizing harmful free radicals through the process of detoxification [80], and environmental stress can enhance the secretion of these scavenging molecules [81]. According to Steward et al. (2000), the selection of a certain cultivar significantly influenced the overall quantity of phenolic compounds in tomatoes, even when they were cultivated in equal environmental conditions [82]. Moreover, the presence of water can significantly impact the process of plant phenolic metabolism and the overall composition of fruit [83]. Conflicting data on the impact of salinity on phenolic compounds in tomato fruits may be found in the literature, which leads to an uncertain increase [84,85], decrease [86], or perhaps maintenance at the same level [87]. Similarly, researchers have observed that tomato fruits grown in saline conditions have an elevated level of flavonoids, indicating an increase in the overall flavonoid content [86], and others confirming a decrease [88]. Rutin, a flavonoid belonging to the category of phenolic chemicals, has a wide range of significant biological and pharmacological effects [89]. Growers must comprehend the delicate equilibrium between optimizing phytochemical content through irrigation regulation while ensuring product quality remains high. The study by Sánchez-Rodríguez et al. (2012) indicated that tomato drought-tolerant cv. Zarina-like rootstocks (ZarxJos) have a better-quality value and are a potential source of health-promoting chemicals due to their abundance of bioactive compounds, especially under water stress situations [90]. ZnO-NPs have been found to enhance nutrient absorption, regulate the Na^+^/K^+^ ratio, maintain water balance, enable ion accumulation, and mitigate the negative impacts of abiotic stressors. The increase in flavonoid, anthocyanin, phenolic, and photosynthetic pigment levels, as well as the upregulation of antioxidant enzymes, are the factors that contribute to these advantages [7].

### 3.4. Determination of Some Biochemical and Stress Markers in the Leaves from Different Treatments of Tomato

The increase in H_2_O_2_ (a reactive oxygen substance that increases due to stress and is highly destructive), MDA (a cytotoxin produced by the breakdown of lipid cell membranes due to ROS), proline (an amino acid that increases in reaction to stress and functions as an effective osmolyte), and total hydrolyzable sugar contents were quite evident under salt stress [91,92,93] (Table 8). The tomato leaves reached their maximum content of total hydrolyzable sugars at the level of 155.97 ± 0.90 µg/g in the fourth treatment (T4), where a concentration of zero mg/L of ZnO-NPs was applied (Table 8). That value was higher than that seen in the control group (T1) and the other two non-stressed treatments (T2 and T3) in concentrations of 84.58 ± 4.10, 107.25 ± 3.43, and 121.78 ± 4.70 µg/g, respectively. Another increased content of sugars was observed in the fifth and sixth stressed treatments with 103.22 ± 4.04 and 121.83 ± 4.04 µg/g, respectively. Applying ZnO-NPs (75 and 150 mg/L) on the leaves of the fifth and sixth treatments (T5 and T6) decreased the sugar content against the severe NaCl concentration (150 mM). The same manner was observed in the proline, H_2_O_2_, and MDA contents. The tomato leaves showed the highest contents of those stress markers in the fourth treatment with 71.54 ± 2.60 µg/g, 1158.76 ± 11.40 µg/g, and 11.77 ± 0.24 mmols/mL, respectively. The content was decreased in the fifth and sixth treatments after spraying with 75 and 150 mg/L ZnO-NPs to be 45.67 ± 1.55 and 41.38 ± 1.91 µg/g for proline, 682.57 ± 4.67 and 558.76 ± 7.38 for H_2_O_2_, and 2.36 ± 0.08 and 3.55 ± 0.06 mmols/mL for MDA, respectively. The contents of tomato leaves from proline, H_2_O_2_, and MDA in the non-stressed treatments (T1–T3) were less than in the stressed treatments (T4–T6) (Table 8). The reason for the increase in MDA and H_2_O_2_ in plants under salt stress is due to the increase of reactive oxygen species (ROS) and cell membrane damage with stress [7]. In a separate study on tomatoes, Li (2009) discovered that the salinity caused an increase in the levels of proline and MDA in tomato seedlings [94]. Furthermore, the amount of NaCl had an impact on the soluble sugar content, as Shahba et al. discovered that the levels of soluble sugars and proline in tomatoes rise in response to increased salt [95].

The tomato leaves reached their maximum content of total free amino acids content in the fourth treatment (T4) at the level of 986.68 ± 8.61 µg/g (Table 8). That value was higher than that seen in the control group (T1) and the other two non-stressed treatments (T2 and T3) in concentrations of 163.73 ± 2.92 and 265.09 ± 5.63 and 352.59 ± 2.84 µg/g, respectively. Another increased content of free amino acids was observed in the fifth and sixth stressed treatments compared to the control group (T1), but that increase was much less than in the fourth treatment when exposed to 150 mM and did not receive any spray of ZnO-NPs. The tomato leaves reached their maximum content of protein content in the first treatment (T1) at the level of 81.28 ± 1.16 µg/g (Table 8). That value was close to the contents of the second and third treatments (T2 and T3) that were sprayed with 75 and 150 mg/L ZnO-NPs, respectively. The protein content in tomato leaves reached its lowest content in the fourth treatment, was stressed with 150 mM, and did not receive any spray of ZnO-NPs in a concentration of 26.11 ± 0.33 µg/g, but there was an evident increase in the contents of protein in the fifth and sixth treatments (T5 and T6) with concentrations of 37.78 ± 0.41 and 38.89 ± 0.75 µg/g, respectively. The study conducted by Faizan et al. (2021) found that tomato seedlings grown in soil containing NaCl showed a reduction in protein content [13]. The decrease in cellular integrity and excessive formation of reactive oxygen species (ROS) was caused by the presence of NaCl [96]. The protein content significantly increased when ZnO-NPs (10, 50, or 100 mg/L) were applied to the leaves, regardless of the presence or absence of NaCl stress. This indicates that ZnO-NPs had a good effect in reducing the impact of NaCl stress. Tavallali et al. (2009) have previously reported similar findings, demonstrating that Zn^2+^ application can enhance protein content in pistachio plants subjected to NaCl stress [97]. The increase in protein content following the injection of Zn^2+^ may be attributed to the reduction in ion leakage, which, in turn, improves the damage caused by NaCl stress [98].

## 4. Materials and Methods

### 4.1. Design, Preparations, and Equipment for Tomato Planting

Tomato seeds of the variety Kecskemétii 549 (K-549) *Solanum lycopersicum* were used for the current experiment. Healthy tomato seeds were surface sterilized with 70% ethanol for 2 min and washed with deionized water three times. The seeds of the tomato cultivar were sown in a plastic tray (seedling tray) (29 May 2023), and after one month at the 3–4 true leaves stage (27 June 2023), the seedlings were transplanted in individual plastic pots. At 30 days after sowing (DAS), the plants were transferred into 28 cm diameter and 28 cm depth pots packed with soil and peat moss in a ratio of 1:1. Humidity and temperature inside the greenhouse were adjusted using a ventilation system and an automated window opening system at a range of day/night temperatures of 25/20 ± 5 °C and 60–65% relative humidity. Six treatments were applied, namely T1: control treatment (irrigation with distilled water “dw”), T2: irrigation with distilled water + foliar spray of ZnO-NPs (75 mg/L), T3: irrigation distilled water + foliar spray of ZnO-NPs (150 mg/L), T4: irrigation with saline solutions (150 mM NaCl), T5: irrigation with saline solutions (150 mM NaCl) + foliar spray of ZnO-NPs (75 mg/L), and T6: irrigation with saline solutions (150 mM NaCl) + foliar spray of ZnO-NPs (150 mg/L). The NaCl (150 mM) stress solution was applied in the soil on the 10th day after the transplanting (40 DAS) to provide salt stress. A foliar spray of ZnO-NPs was applied at different concentrations *viz.*, 75 or 150 mg/L three times at 10 days after salt stress (20, 34, and 48 DAT) (50, 64, and 78 DAS), and these specific times were chosen to include the vegetative and generative phases in the tomato plants. Control plants were treated with only distilled water. To unify soil electric conductivity (EC), EC water drainage was monitored daily for all treatments, and it was maintained equal to the EC irrigating solution. That was done by allowing enough drainage from the root zone by the use of a corresponding saline solution until equilibrium between the EC of water drainage and irrigating solution. In that way, the EC root zone stabilized at the specified set point during the experiment. Consequently, the amount of water added at each time ranged between 1 and 1.5 Liters. It was adjusted based on the EC water drainage obtained at each time. Plant height, stem diameter, leaf area, and chlorophyll content were measured first at 10 days after transplanting (DAT). Moreover, 2 kg of pre-washed and dried gravel was used to line the pot base and was covered by a plastic net. Three 8 mm holes were designed at the bottom of the pot using a hole-punched device for watering and aeration purposes. The remaining empty pot was filled with soil peat mixture (1:1 by weight): soil from “A” horizon of an Eutric cambisol soil, having a sandy clay loam texture, was collected from the research farm area of the Hungarian University of Agriculture and Life Sciences, Georgikon Campus, and Baltic peat (DURPETA FÖLD PH 5,5-6,5, profi mix 2B 250 L, 04789) was obtained from Latvia. Both soil and peat were sieved through a 4 mm sieve to obtain a finer and more favorable growth medium. Then, 4.5 kg of soil and 4.5 kg of peat were mixed using a cement mixer to obtain a homogenized mixture that was used as a growth medium in pots. The moisture content and water-holding capacity of the soil peat mixture were determined by the gravimetric method [99] to quantify the amount of irrigation to be supplied to control and treated pots.

### 4.2. Determining the Moisture Content of the Potting Soil

Air-dried soil (soil + peat moss 1:1), even when thoroughly air-dried, always has moisture, no matter how little. This moisture had to be accounted for when determining the amount of water needed to bring soil to its pot capacity. The first step thus involved the determination of the potting soil’s moisture content, i.e., its gravimetric water content.

### 4.3. Determining the Amount of Air-Dried Soil to Place in Each Pot

Each pot was filled with an amount of soil that would give the required amount of oven-dried soil. Overall, 8445.9 g of oven-dried soil was needed since the air-dried soil had a moisture content of 6.56% (0.0656 g/g). Then, the amount of air-dried soil needed to give a mass of 8445.9 g (8.4459 kg) of oven-dried soil was calculated by determining the amount of water needed to bring a mass of potting soil to pot capacity. The pot capacity (moisture content of the pot) was 67.15%. That value constituted the full amount of water needed to bring the soil to 100% pot capacity. However, a certain mass of air-dried soil was added to each pot, and that air-dried soil was an equivalent mass of oven-dried soil. It was also taken into account that the air-dried soil contained moisture. That amount of moisture needed to be accounted for no matter how small it might seem to be. The moisture needed to bring the soil to pot capacity was 67.15% (0.6715 g/g). This means that each gram of soil would need 0.6715 g of moisture to bring it to pot capacity. However, the air-dried soil sample already contained 0.0656% moisture. The amount of water needed to bring this soil to pot capacity was as follows: The water needed to bring air-dried soil to 100% pot capacity (PC) = moisture content at PC − moisture content in air-dried soil = 67.15% − 6.56% = 60.59% or (0.6059 g water/g oven-dried soil). The water needed to bring 10 kg oven-dried soil to 100% PC = (0.6059 g of water × 8445.9 g of soil)/1 g = 5117.37 g (5117.37 mL) (5.117 L) (5 L and 117 mL). Since water is much easier to add in volumes than as a weight, the density of water, which is 1 g/cm^3^, was recalled. The volume of water needed to be added to the air-dried mass of potting soil of 9 kg to bring it to pot capacity (100%) was already known. The 100% pot capacity would keep plants in pots under well-watered conditions. The equation (y = 1.0432x + 0.0569) of the calibration curve of the time domain reflectometer (TDR; FieldScout TDR 300 Soil Moisture Meter, Spectrum Technologies, Inc., Aurora, IL, USA) was used to determine the water requirements every week to irrigate the plants during the whole experiment. The value on the device is substituted in the equation. The result was subtracted from 0.66, which was the moisture of the pot capacity. After the amount of moisture that must be added to make the soils in pots around the pot capacity was calculated, it was multiplied by 8445, which was the weight of completely oven-dried soil. The water requirements were recorded regularly.

### 4.4. Determination of the Macro and Microelements in the Leaves of Tomato Plants

The leaf samples were dried in an oven at 70 °C for 24 h. Approximately 0.2 g of the dried leaf and bark samples, respectively, were treated individually with 8 mL HNO_3_ (65% Merck) and 10 mL H_2_O_2_ (30% Merck). Then, they were mineralized using a microwave digestion system for 40 min. After 40 min of digestion, the samples were cooled for 30 min, and the clear solutions were filtered and brought at 50 mL with distilled deionized water. Metals concentrations in the final solutions were analyzed by Flame Atomic Absorption Spectrometry (FAAS; iCE 3000 Series, Thermo Fisher Scientific Inc., Waltham, MA, USA). The concentration of the primary standards of Na, K, Mg, Zn, Cu, and Mn elements was 1000 ppm (1000 mg/L), and the matrixes were 2% HNO_3_ [100].

### 4.5. Determination of Chlorophyll and Growth Attributes

Chlorophyll content per unit area was determined using Soil Plant Analysis Development (SPAD) 502 Plus, Konica Minolta. The readings were recorded in SPAD units. SPAD values were recorded according to the biggest leaf on all its leaflets. The readings were recorded every 10 days starting at 10 DAT. The area of tomato leaf (cm^2^) was determined according to [101]. It was recorded using the following equation: S = L.W.K. The maximum width (W) and length (L) of each sampled leaf were measured with a ruler. In tomato, the length was measured as the distance between the insertion of the first leaflet on the rachis to the distal end, whereas the width was measured on the widest leaflet. S: leaf surface area of the plant (cm^2^). L: maximum leaf length (cm). W: maximum leaf width (cm). The correction coefficient (K) for the tomato surface was 0.5. Tomato plant height was measured in centimeters at each plant from the soil surface using the ruler that was placed on the ground next to the stem and measured to the height of the tallest stem (ignoring the leaves). The stem thickness of each plant was measured in centimeters using the caliber. In the current study, the previous recordings were collected in 4 different stages.

### 4.6. Determination of Proline and Protein Contents

Fresh 500 mg leaf samples were extracted with sulfosalicylic acid. The extraction also used equal amounts of glacial acetic acid and acidic ninhydrin. After heating at 100 °C, 4 mL of toluene was added to the samples. A Genesys (PG Instruments Ltd., T 80, Lutterworth, UK) spectrophotometer evaluated the aspirated layer’s absorbance at 520 nm. Proline concentration was measured in micrograms per gram FW [102]. To determine the protein content, a homogenized 1 g of newly generated leaves was added into a protein extraction buffer. The buffer contained 10 mM tris-HCl (pH = 8.1), 5 mM β-mercaptoethanol, 0.57 mM PMSF, and 10 mM pH 8 EDTA. The mixture was then centrifuged for 10 min at 17,709× *g*. After gathering the liquid, Bradford reagent was used to induce the color. A Genesys (PG Instruments Ltd., T 80, UK) spectrophotometer measured the color intensity at 595 nm, and the protein concentration was presented in micrograms per gram FW [103].

### 4.7. Determination of Total Free Amino Acids

The fresh leaves were pulverized into a fine powder using a mortar and pestle. The resulting powder was then mixed with 10 mL of phosphate buffer solution, which had a concentration of 0.05 M. The pulverized leaves were centrifuged at 12,298× *g* for 10 min at 4 °C. The reaction mixture in a test tube contained 0.5 mL of the extract, 0.5 mL of 4% ninhydrin, and 0.5 mL of 2% pyridine. The test tubes were vigorously vortexed and heated at 100 °C for 30 min in a water bath. The optical density at 570 nm was measured using a spectrophotometer (Genesys PG Instruments Ltd., T 80, UK) [104].

### 4.8. Determination of Total Hydrolyzable Sugars

The total soluble sugars were extracted and estimated using a significantly modified method. Dry tissue was immersed in 5 mL of 2.5 N HCl in a boiling water bath for 3 h with agitation to extract soluble sugars. Finally, the samples were centrifuged at 3075× *g*. The extracts were boiled for 10 min with 3.0 mL of freshly prepared anthrone and chilled to quantify total soluble sugars. The spectrophotometer (Genesys PG Instruments Ltd., T 80, UK) measured the green color intensity at 630 nm absorbance [105].

### 4.9. Determination of Lipid Peroxidation

A slightly modified method used malondialdehyde (MDA) concentration to quantify lipid peroxidation. During this process, 100 mg of fresh plant leaves were well mixed in 1 mL of 0.1% TCA. After vigorous mixing, the tubes were centrifuged at 20,784× *g* at 4 °C for 10 min. In a Falcon tube, 800 microliters of centrifuged liquid were mixed with 2 milliliters of 0.5% TBA solution. A water bath heated the mixture to 80 °C for 25 min. After incubation, the tubes were chilled on ice for 5 min and centrifuged at 20,784× *g* at 4 °C for 10 min to separate and collect any remaining TBA. Using a Genesys (PG Instruments Ltd., T 80, UK) spectrophotometer at 532 and 600 nm, optical density was determined [106].

### 4.10. Determination of Hydrogen Peroxide (H_2_O_2_)

The fresh leaves, weighing approximately 0.5 g, were crushed in 5 mL of a solution containing 0.1% (*w*/*w*) trichloroacetic acid. The resulting mixture was centrifuged at 17,709× *g* for 15 min. Phosphate buffer with a pH of 7.2 and a concentration of 100 mM, and potassium iodide were added into a 0.5 mL solution of the reaction mixture. The achieved color was measured using a UV-visible spectrophotometer (Genesys PG Instruments Ltd., T 80, UK), which measured the absorbance of the mixture at a wavelength of 390 nm [107].

### 4.11. Determination of Total Flavonoid Compounds (TFCs)

The flavonoid levels in tomato aqueous extracts were assessed using the aluminum chloride colorimetric technique. Test tubes filled with tomato leaf extract were mixed with 5% sodium nitrite, 10% aluminum chloride, and 1 M sodium hydroxide. The tubes were incubated for 15 min, and the absorbance of the pink color was measured using a spectrophotometer (Genesys PG Instruments Ltd., T 80, UK) at 510 nm. Quercetin was used as a standard substance, and the total flavonoid content was quantified as the amount of Quercetin equivalent in micrograms per gram of dry extract [108].

### 4.12. Determination of Total Phenolic Compounds (TPCs)

The phenolic content of tomato leaves was quantified using the Folin–Ciocalteu reagent method. The extract was adjusted to 2500 µg/mL and added to a solution of 1 N Folin–Ciocalteu reagent and 20% sodium carbonate. The mixture was kept in a dark environment for 90 min. The blue color was measured at 650 nm using a spectrophotometer (Genesys PG Instruments Ltd., T 80, UK), and the total phenolic content was determined by calculating the equivalent amount of Gallic acid in micrograms per gram of dry extract [108].

### 4.13. Determination of Phenolic Compounds; High-Performance Liquid Chromatography (HPLC) Analysis

The powder sample was dissolved in 2 M sodium hydroxide, flushed with N_2_, and stopped. The pH was adjusted to 2 with 6 M hydrochloric acid. The resulting supernatant was collected, and phenolic compounds were extracted twice with 50 mL of ethyl ether and ethyl acetate. The organic phase was separated, and the solvent evaporated at 45 °C. The residues were re-dissolved in methanol and analyzed using HPLC. The analysis was conducted at the National Research Centre, Dokki, Cairo, Egypt [109]. The analytical column used was an Eclipse XDB-C18 with a C18 guard column (Phenomenex, Torrance, CA, USA), and the mobile phase was acetonitrile and 2% acetic acid in water. The flow rate was maintained at 0.8 mL/min for 60 min, with a gradient program from 100% B to 85% B in 30 min, from 85% B to 50% B in 20 min, from 50% B to 0% B in 5 min, and from 0% B to 100% B in 5 min. The injection volume was 50 µL, and peaks were monitored at 280, 320, and 360 nm for benzoic acid, cinnamic acid derivatives, and flavonoids. All samples were filtered through a 0.45 µm Acrodisc syringe filter (Gelman Laboratory, MI) before injection. Peaks were identified by congruent retention times and UV spectra and compared with those of the standards.

### 4.14. Chemical Synthesis of Zinc Oxide Nanoparticles

ZnO nanoparticles were synthesized by the precipitation method using zinc nitrate hexahydrate “Zn(NO_3_)_2_⋅6H_2_O, thermos scientific A16282, LOT: 10242486” and sodium hydroxide “NaOH, 98.5–100.5%, pellets, AnalaR NORMAPUR^®^ Reag. Ph. Eur. analytical reagent” as precursors. ZnO nanoparticles were produced by mixing aqueous solutions of zinc nitrate and sodium hydrate. ZnO nanoparticles were formed by the reaction between Zn^2+^ and hydroxide ions as shown by the following equations [110]; (1) Zn(NO_3_)_2_→Zn^2+^ + 2NO^3−^ (2) Zn^2+^ + 2OH^−^→Zn(OH)_2_ (3) Zn(OH)_2_→ZnO + H_2_O. The aqueous solution was prepared by mixing zinc nitrate hexahydrate and sodium hydroxide. The process involved dissolving 2.28 g of zinc nitrate hexahydrate in 75 mL of pure water, then adding 0.6 g of NaOH in 150 mL of pure water dropwise under magnetic stirring. The solution was centrifuged at 7871× *g* for 10 min at 25 °C, washed with pure water, dried at 60 °C for 24 h, and calcined at 200 °C for 2 h in an oven. Different concentrations of ZnO-nanoparticles solution for foliar application were prepared by dissolving the obtained white crystals in distilled water using ultrasonic waves for 10 min.

### 4.15. Characterization of Chemically Synthesized Zinc Oxide Nanoparticles

The optical properties of ZnO-NPs were monitored by UV–VIS absorption spectroscopy by Genesys spectrophotometer (PG Instruments Ltd., T 80, UK) in the central lab of the Agronomy Department, Hungarian University of Agriculture and Life Sciences, Georgikon campus, Hungary, Keszthely. UV–VIS spectra were recorded in the wavelength range of 300–700 nm [28]. XRD measurements were carried out at the Micro Analytical Center, Cairo University, Faculty of Science, Giza, Egypt, using a Philips PW 3710-type instrument (CuKα radiation, *λ*  =  1.54056 Å, 50 kV, 40 mA), in the 4°–15° 2*θ* range with a scanning speed of 0.02°/s and 1 s dwell time. The finely powdered samples were loaded into back-packed mounts to minimize preferential particle orientation. Calcined ZnO-NPs were applied as a reference sample for estimating the instrumental broadening for the average crystallite size calculation [28]. FTIR spectra were recorded on a Bruker Vertex 70v type equipment with a Bruker Diamond ATR compartment with a resolution of 2 cm^−1^ using a DTGS detector at Nanolab, Pannon University, Veszprém, Hungary. The recorded spectra were the average of 512 individual scans [28]. Transmission electron microscope investigations were carried out at the Micro Analytical Center, Cairo University, Faculty of Science, Giza, Egypt, using a FEI Talos F200X type electron microscope Thermo Fisher Scientific Inc., with an X-FEG electron source, operated at 200 kV accelerator potential, in conventional transmission (TEM) and scanning modes (STEM). The composition of samples was analyzed using an energy-dispersive X-ray (EDAX) spectrometer attached to the transmission electron microscope. Both scanning and conventional TEM images were recorded. The samples were prepared from aqueous dispersion (MilliQ) by laying a droplet onto a lacy carbon-coated copper grid, which was dried at 60 °C before measurement [111]. The hydrodynamic size and Zeta potential of ZnO-NPs in both water and full cell culture media were measured using dynamic light scattering (DLS) with the Nano-ZetaSizer-HT device from Malvern Instruments Zetasizer Nano-ZS in the UK. DLS examines the velocity distribution of particle motion by quantifying dynamic variations in light scattering intensity resulting from the Brownian motion of the particle. This method produces a hydrodynamic radius or diameter, which can be determined using the Stokes–Einstein equation. It provides a comprehensive measurement of the particle’s perpendicular dimension to the light source at that particular moment. The Zetasizer Nano-ZS utilizes laser Doppler velocimetry (LDV) as its measurement technique to determine the Zeta potential of particles in a solution. This method employs a laser that is directed into the sample to determine the speed of the particles under the influence of a known electric field, referred to as electrophoretic mobility. The device utilizes a 4 mW He-Ne 633 nm laser to analyze the samples, in addition to an electric field generator specifically designed for LDV measurements. The ZnO-NPs were diluted in deionized water and mixed vigorously for 5 min to create a uniform solution. Then, 1 mL of the solution was transferred to a Malvern Clear Zeta Potential cell for LDV measurements. The majority of solutions were prepared at a concentration of 25 µg/mL [112].

### 4.16. Statistical Analyzsis

The experimental techniques in this study consisted of performing all assessments four times, and the resulting results were reported as the average value together with the standard error. The statistical analysis was performed utilizing the Web Agri Stat Package (WASP) at the ICAR: Central Coastal Agricultural Research Institute. The study utilized a one-way analysis of variance (ANOVA) to investigate differences between groups. The least significant difference (LSD) test was employed using significance thresholds of 1% and 5% [113].

## 5. Conclusions

Abiotic stress conditions impede worldwide agricultural production. Salt disrupts the ecological balance of the region and reduces the productivity of most crops. Increased salinity reduces the amount of water that plants can obtain. It also affects the physicochemical qualities of the ground. Salt diminishes agricultural yield, degrades the soil, and adversely impacts the economy. Saline impacts arise from intricate interactions among morphological, physiological, and biochemical processes, encompassing seed germination, plant growth, and the uptake of water and nutrients. Plant growth is negatively affected by early salt stress. This ultimately decreases crop output in saline environments, affecting both the quality and quantity of plant products. This underscores the necessity of researching procedures involving the utilization of manufactured nanoparticles to govern plants’ physiological responses to adverse environments.

The present study demonstrated the mitigating impact of chemically produced zinc oxide nanoparticles and their ability to counteract the detrimental effects of the 150 mM NaCl stressor. This study aimed to assess the impact of chemically generated ZnO-NPs foliar spray at several concentrations on tomato plants under salt stress. The results were analyzed in terms of numerous morphological and biochemical factors. A notable decline in plant growth characteristics and development was seen in correlation with the salt stress. Applying ZnO-NPs externally to the leaves enhanced the ability of tomato plants to withstand salt stress. Zinc oxide nanoparticles (ZnO-NPs) play a protective role in tomato plants exposed to salt stress. They help safeguard the integrity of the plant’s cell membranes, enhance chlorophyll content, increase the accumulation of polyphenolic compounds, induce changes in biochemical stress indicators, and regulate mineral levels. Therefore, due to the therapeutic properties of ZnO-NPs in tomato plants exposed to salt stress, it can be regarded as a viable alternative for cultivation in regions with limited access to high-quality irrigation water. Additionally, it would be beneficial to conduct further research on the impacts at the enzyme and molecular levels to obtain more comprehensive and precise findings. Furthermore, as a furtherance of the investigation, it is possible to undertake studies that extend to the point of fruit yield and ripening in plants to optimize the dosages of zinc oxide nanoparticles applied through foliar spray.

## Figures and Tables

**Figure 1 plants-13-01418-f001:**
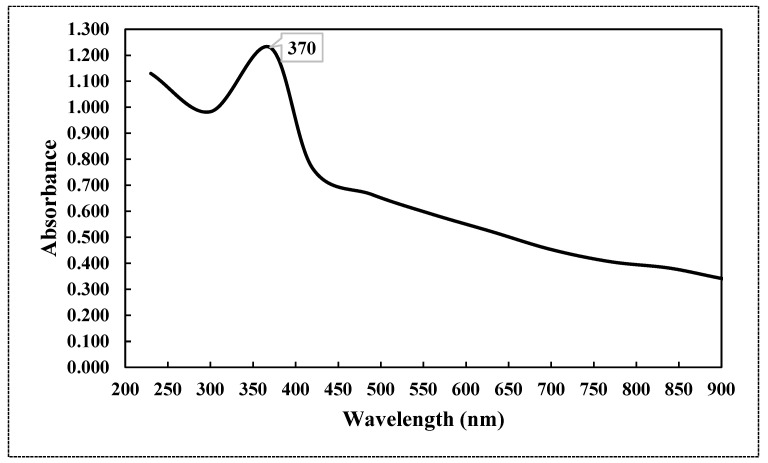
UV.VIS spectrum of chemically synthesized ZnO-NPs.

**Figure 2 plants-13-01418-f002:**
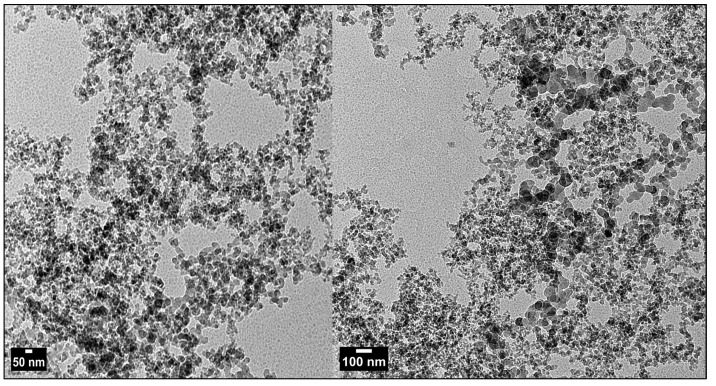
Electron micrographs (TEM) of chemically synthesized ZnO-NPs on 50 and 100 nm.

**Figure 3 plants-13-01418-f003:**
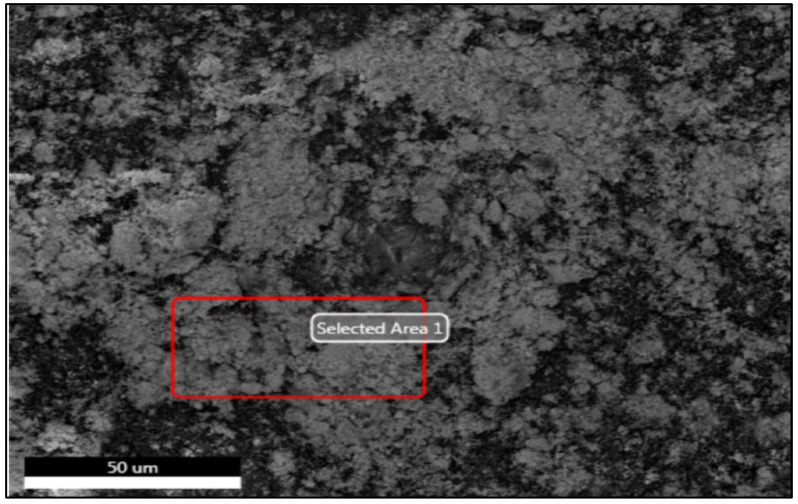
Electron micrographs (STEM) of chemically synthesized ZnO-NPs on 50 microns.

**Figure 4 plants-13-01418-f004:**
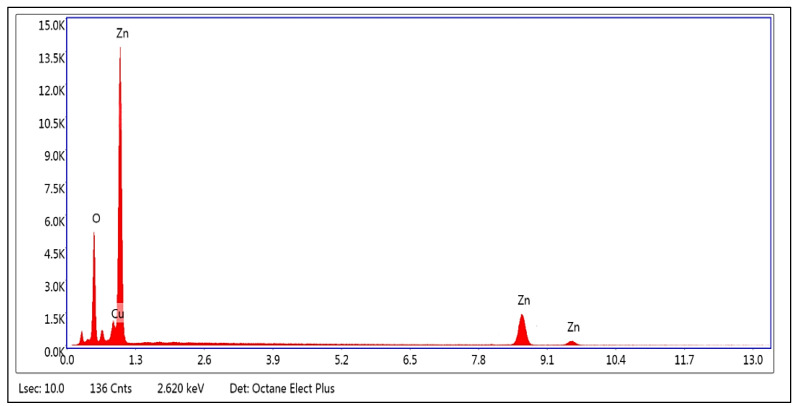
EDAX spectrum of chemically synthesized ZnO-NPs.

**Figure 5 plants-13-01418-f005:**
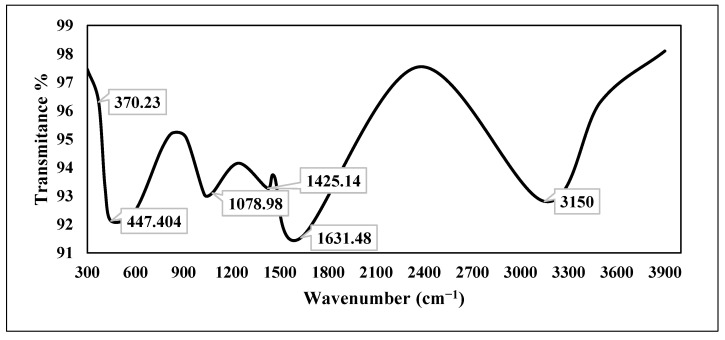
FTIR spectrum of chemically synthesized ZnO-NPs.

**Figure 6 plants-13-01418-f006:**
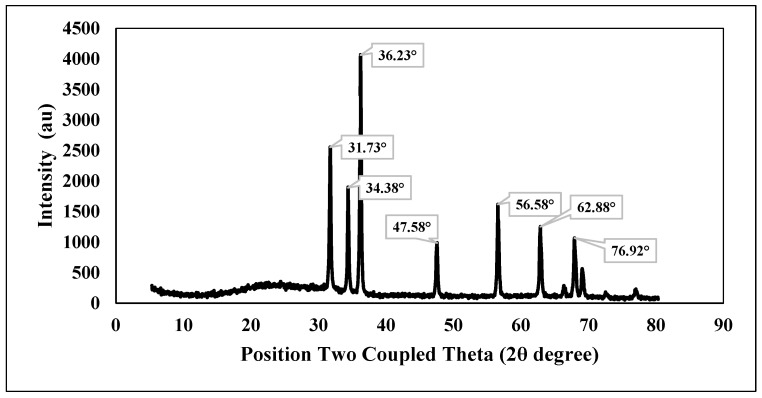
XRD pattern of chemically synthesized ZnO-NPs.

**Figure 7 plants-13-01418-f007:**
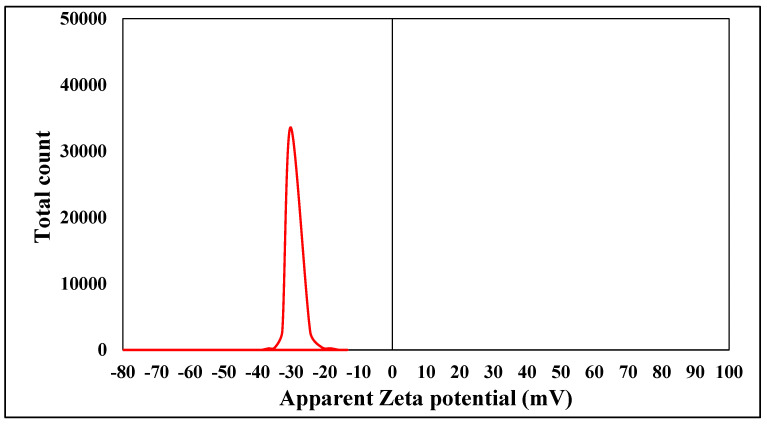
Zeta potential of chemically synthesized ZnO-NPs measuring the charge.

**Table 1 plants-13-01418-t001:** eZAF Smart Quant results and oxide results.

Element	Weight %	Atomic %
ZnK	67.55	42.95
OxygenK	32.02	56.81
CuK	0.43	0.23

**Table 2 plants-13-01418-t002:** Determination of some macro and microelements in the leaves of tomato.

Treatments	Concentrations (µg/g)					
	Na	K	Mg	Zn	Cu	Mn
T1 Control (dw)	1.47 ± 0.13 ^c^	29.20 ± 0.22 ^a^	5.57 ± 0.02 ^ns^	0.48 ± 0.11 ^c^	0.01 ± 0.04 ^c^	0.22 ± 0.01 ^ns^
T2 (dw+ZnO-NPs 75 mg/L)	1.31 ± 0.10 ^c^	29.02 ± 0.02 ^a^	5.56 ± 0.02 ^ns^	2.72 ± 0.37 ^b^	0.02 ± 0.03 ^c^	0.22 ± 0.00 ^ns^
T3 (dw+ZnO-NPs 150 mg/L)	1.49 ± 0.06 ^c^	29.10 ± 0.23 ^a^	5.59 ± 0.02 ^ns^	3.06 ± 0.58 ^ab^	0.01 ± 0.01 ^c^	0.21 ± 0.01 ^ns^
T4 NaCl (150 mM)	26.48 ± 0.21 ^a^	13.56 ± 0.38 ^c^	5.63 ± 0.03 ^ns^	0.47 ± 0.08 ^c^	0.02 ± 0.03 ^b^	0.63 ± 0.00 ^ns^
T5 (150 mM NaCl+ZnO-NPs 75 mg/L)	13.51 ± 0.06 ^b^	26.66 ± 0.40 ^b^	5.63 ± 0.01 ^ns^	2.44 ± 0.18 ^b^	0.004 ± 0.09 ^ab^	0.77 ± 0.00 ^ns^
T6 (150 mM NaCl+ZnO-NPs 150 mg/L)	13.27 ± 0.23 ^b^	26.34 ± 0.42 ^b^	5.62 ± 0.01 ^ns^	4.13 ± 0.38 ^a^	0.012 ± 0.05 ^a^	0.78 ± 0.00 ^ns^
LSD _(0.01)_	0.606	1.267	ND	1.692	0.204	ND
LSD _(0.05)_	0.442	0.925	ND	1.235	0.149	ND
Coefficient of variation	4.010	2.431	0.684	40.099	21.315	59.121

Each value represents the mean ± SE. Values with the same letter are not significantly different at (*p* ≤ 0.05), and the comparison is done according to different treatments in the same column. ND: non-determined.

**Table 3 plants-13-01418-t003:** Determination of chlorophyll content (SPAD-units) in the leaves of tomato.

Treatments	Readings of Chlorophyll Content in Different Stages			
	5 July 2023	19 July 2023	28 July 2023	11 August 2023
T1 Control (dw)	48.50 ± 1.89 ^ns^	51.98 ± 2.82 ^ns^	60.43 ± 1.24 ^ab^	63.13 ± 1.97 ^a^
T2 (dw+ZnO-NPs 75 mg/L)	46.05 ± 0.42 ^ns^	57.78 ± 1.65 ^ns^	60.13 ± 2.55 ^b^	61.25 ± 2.25 ^ab^
T3 (dw+ZnO-NPs 150 mg/L)	44.60 ± 1.06 ^ns^	56.83 ± 1.05 ^ns^	62.75 ± 2.09 ^a^	63.75 ± 2.43 ^a^
T4 NaCl (150 mM)	46.93 ± 0.93 ^ns^	54.88 ± 1.14 ^ns^	52.65 ± 0.44 ^cd^	55.25 ± 0.85 ^c^
T5 (150 mM NaCl+ZnO-NPs 75 mg/L)	49.20 ± 0.71 ^ns^	55.78 ± 1.05 ^ns^	53.18 ± 2.22 ^c^	60.75 ± 1.11 ^abc^
T6 (150 mM NaCl+ZnO-NPs 150 mg/L)	48.68 ± 1.08 ^ns^	54.68 ± 2.75 ^ns^	52.48 ± 1.26 ^d^	61.68 ± 2.47 ^ab^
LSD _(0.01)_	ND	ND	ND	ND
LSD _(0.05)_	ND	ND	5.404	5.805
Coefficient of variation	4.696	6.482	6.444	6.531

Each value represents the mean ± SE. Values with the same letter are not significantly different at (*p* ≤ 0.05), and the comparison is done according to different treatments in the same column. ND: non-determined.

**Table 4 plants-13-01418-t004:** Determination of growth attributes (plant height) in tomato.

Treatments	Readings of Plant Height (cm) in Different Stages			
	5 July 2023	19 July 2023	28 July 2023	11 August 2023
T1 Control (dw)	24.80 ± 2.16 ^ns^	47.50 ± 1.19 ^ns^	57.75 ± 1.93 ^b^	60.75 ± 2.01 ^c^
T2 (dw+ZnO-NPs 75 mg/L)	28.38 ± 2.51 ^ns^	50.75 ± 0.85 ^ns^	63.50 ± 2.72 ^a^	67.00 ± 2.16 ^b^
T3 (dw+ZnO-NPs 150 mg/L)	28.38 ± 2.50 ^ns^	50.75 ± 3.48 ^ns^	63.50 ± 1.80 ^a^	75.25 ± 2.10 ^a^
T4 NaCl (150 mM)	25.50 ± 2.25 ^ns^	47.00 ± 0.71 ^ns^	54.75 ± 2.25 ^c^	57.00 ± 2.42 ^d^
T5 (150 mM NaCl+ZnO-NPs 75 mg/L)	25.75 ± 1.65 ^ns^	49.50 ± 1.19 ^ns^	57.75 ± 2.98 ^b^	60.00 ± 1.68 ^cd^
T6 (150 mM NaCl+ZnO-NPs 150 mg/L)	24.13 ± 1.68 ^ns^	47.00 ± 0.58 ^ns^	57.25 ± 2.75 ^bc^	61.13 ± 2.09 ^bc^
LSD _(0.01)_	ND	ND	10.062	8.369
LSD _(0.05)_	ND	ND	7.345	6.110
Coefficient of variation	16.804	6.796	8.116	6.530

Each value represents the mean ± SE to the approximate centimeters. Values with the same letter are not significantly different at (*p* ≤ 0.05), and the comparison is done according to different treatments in the same column. ND: non-determined.

**Table 5 plants-13-01418-t005:** Determination of growth attributes (stem width) in tomato.

Treatments	Readings of Stem Width (cm) in Different Stages			
	5 July 2023	19 July 2023	28 July 2023	11 August 2023
T1 Control (dw)	0.48 ± 0.03 ^ns^	0.73 ± 0.73 ^abc^	0.95 ± 0.03 ^b^	1.13 ± 0.01 ^ab^
T2 (dw+ZnO-NPs 75 mg/L)	0.50 ± 0.00 ^ns^	0.86 ± 0.00 ^a^	1.09 ± 0.08 ^a^	1.20 ± 0.02 ^a^
T3 (dw+ZnO-NPs 150 mg/L)	0.50 ± 0.03 ^ns^	0.86 ± 0.05 ^ab^	1.08 ± 0.03 ^ab^	1.16 ± 0.04 ^ab^
T4 NaCl (150 mM)	0.50 ± 0.00 ^ns^	0.64 ± 0.02 ^bc^	0.75 ± 0.04 ^d^	0.78 ± 0.03 ^d^
T5 (150 mM NaCl+ZnO-NPs 75 mg/L)	0.48 ± 0.03 ^ns^	0.59 ± 0.01 ^c^	0.81 ± 0.02 ^c^	0.90 ± 0.02 ^c^
T6 (150 mM NaCl+ZnO-NPs 150 mg/L)	0.46 ± 0.02 ^ns^	0.66 ± 0.02 ^bc^	0.94 ± 0.04 ^b^	1.09 ± 0.05 ^b^
LSD _(0.01)_	ND	0.190	0.139	0.126
LSD _(0.05)_	ND	0.139	0.101	0.092
Coefficient of variation	8.394	13.192	7.244	5.794

Each value represents the mean ± SE to the approximate centimeters. Values with the same letter are not significantly different at (*p* ≤ 0.05), and the comparison is done according to different treatments in the same column. ND: non-determined.

**Table 6 plants-13-01418-t006:** Determination of growth attributes (leaf area) in tomato.

Treatments	Readings of Leaf Area (cm^2^) in Different Stages			
	5 July 2023	19 July 2023	28 July 2023	11 August 2023
T1 Control (dw)	119.56 ± 3.94 ^c^	448.15 ± 15.85 ^a^	510.75 ± 0.75 ^b^	596.10 ± 15.90 ^ab^
T2 (dw+ZnO-NPs 75 mg/L)	182.63 ± 1.38 ^a^	434.73 ± 0.78 ^a^	482.13 ± 3.63 ^bc^	638.10 ± 5.40 ^a^
T3 (dw+ZnO-NPs 150 mg/L)	136.38 ± 1.38 ^bc^	414.63 ± 5.13 ^a^	594.00 ± 16.50 ^a^	632.15 ± 17.15 ^a^
T4 NaCl (150 mM)	154.875 ± 18.38 ^ab^	351.50 ± 26.50 ^b^	392.25 ± 20.25 ^d^	452.75 ± 40.25 ^c^
T5 (150 mM NaCl+ZnO-NPs 75 mg/L)	131.63 ± 1.63 ^bc^	360.88 ± 3.13 ^b^	442.25 ± 7.25 ^cd^	510.80 ± 58.20 ^bc^
T6 (150 mM NaCl+ZnO-NPs 150 mg/L)	136.13 ± 16.13 ^bc^	435.19 ± 14.81 ^a^	479.75 ± 31.75 ^bc^	484.50 ± 34.50 ^bc^
LSD _(0.01)_	ND	ND	89.702	ND
LSD _(0.05)_	35.166	49.134	59.213	116.275
Coefficient of variation	10.012	4.927	5.005	8.602

Each value represents the mean ± SE. Values with the same letter are not significantly different at (*p* ≤ 0.05), and the comparison is done according to different treatments in the same column. ND: non-determined.

**Table 7 plants-13-01418-t007:** HPLC analysis of phenolic compounds in the leaves from different treatments of tomato.

Compounds to Be Detected	RT (min)	Concentration (μg/g)/Treatment					
		T1	T2	T3	T4	T5	T6
Gallic acid	3.7	122.91	149.63	141.73	206.62	192.26	203.61
Protocatechuic acid	6.4	1.38	1.74	18.07	20.26	21.04	24.13
Gentisic acid	9.7	0.00	0.00	0.00	0.00	0.00	0.00
*p*-hydroxybenzoic acid	9.8	10.19	12.96	15.21	19.03	22.25	25.36
Catechin	11.8	25.14	25.23	31.37	95.52	91.60	99.66
Chlorogenic acid	12.7	154.08	239.25	355.54	359.51	617.96	603.38
Caffeic acid	13.5	0.63	2.32	1.59	7.90	5.05	8.69
Syringic acid	14.6	8.10	4.66	44.27	49.33	46.87	46.72
Vanillic acid	16.0	4.36	4.49	4.79	7.74	10.47	9.20
Ferulic acid	20.6	37.35	41.80	44.71	58.35	67.70	80.53
Sinapic acid	21.5	18.88	25.99	36.12	27.67	65.09	60.21
Rutin	24.5	5.58	17.72	1.03	2.11	3.14	5.27
*p*-coumaric acid	25.4	13.22	11.35	0.86	1.13	2.97	27.21
Apigenin-7-glucoside	27.5	5.69	15.31	36.98	52.80	39.21	69.89
Rosmarinic acid	29	0.00	0.00	0.00	0.00	0.00	0.00
Cinnamic acid	35.1	1.61	1.26	7.31	13.80	15.03	23.22
Quercetin	36.3	1.10	2.93	1.02	1.40	2.04	2.71
Apigenin	39.2	0.19	1.19	5.50	4.62	6.79	5.20
Kaempferol	40.8	0.00	0.00	0.00	0.00	0.00	0.00
Chrysin	51.5	4.00	4.19	6.92	9.83	14.31	14.79
Total		414.41	562.02	753.02	937.62	1223.78	1309.78

T1: control (dw); T2: dw+ZnO-NPs 75 mg/L; T3: dw+ZnO-NPs 150 mg/L; T4: NaCl (150 mM); T5: 150 mM NaCl+ZnO-NPs 75 mg/L; and T6: 150 mM NaCl+ZnO-NPs 150 mg/L.

**Table 8 plants-13-01418-t008:** Determination of the different biochemical and stress markers in the leaves from different treatments of tomato.

Treatments	Concentrations							
	TPCs (µg/g)	TFCs (µg/g)	Total Hydrolazable Sugars (µg/g)	Total Free Amino Acids (µg/g)	Protein Content (µg/g)	Proline Content (µg/g)	H_2_O_2_ (µg/g)	MDA (mmols/mL)
T1 Control (dw)	2096 ± 0.10 ^d^	401 ± 3.39 ^b^	84.58 ± 4.10 ^d^	163.73 ± 2.92 ^f^	81.28 ± 1.16 ^a^	15.23 ± 0.64 ^f^	418.76 ± 1.78 ^e^	1.56 ± 0.077 ^e^
T2 (dw+ZnO-NPs 75 mg/L)	2829 ± 0.39 ^bc^	356.56 ± 37.23 ^b^	107.25 ± 3.43 ^c^	265.09 ± 5.63 ^e^	79.11 ± 0.50 ^ab^	26.90 ± 0.42 ^e^	571.14 ± 13.33 ^d^	2.00 ± 0.75 ^d^
T3 (dw+ZnO-NPs 150 mg/L)	3628 ± 0.50 ^ab^	362.48 ± 25.98 ^b^	121.78 ± 4.70 ^b^	352.59 ± 2.84 ^d^	78.61 ± 1.13 ^b^	30.71 ± 1.99 ^d^	744.00 ± 2.33 ^b^	2.41 ± 0.094 ^c^
T4 NaCl (150 mM)	4043 ± 0.44 ^a^	1191.83 ± 16.66 ^a^	155.97 ± 0.90 ^a^	986.68 ± 8.61 ^a^	26.11 ± 0.33 ^d^	71.54 ± 2.60 ^a^	1158.76 ± 11.40 ^a^	11.77 ± 0.24 ^a^
T5 (150 mM NaCl+ZnO-NPs 75 mg/L)	3730 ± 0.32 ^ab^	1222.39 ± 62.84 ^a^	103.22 ± 4.04 ^c^	694.18 ± 7.24 ^b^	37.78 ± 0.41 ^cd^	45.67 ± 1.55 ^b^	682.57 ± 4.67 ^c^	2.37 ± 0.08 ^c^
T6 (150 mM NaCl+ZnO-NPs 150 mg/L)	2596 ± 0.11 ^c^	1271.37 ± 79.35 ^a^	121.83 ± 4.04 ^b^	552.14 ± 3.35 ^c^	38.89 ± 0.75 ^c^	41.38 ± 1.91 ^c^	558.76 ± 7.38 ^d^	3.55 ± 0.06 ^b^
LSD _(0.01)_	1408.995	152.394	9.852	22.406	3.206	4.661	32.934	0.448
LSD _(0.05)_	1028.595	111.251	7.192	16.357	2.341	3.403	24.043	0.327
Coefficient of variation	21.954	9.350	4.182	2.192	2.938	5.938	2.349	5.588

Each value represents the mean ± SE. Values with the same letter are not significantly different at (*p* ≤ 0.05), and the comparison was performed according to different treatments in the same column.

## Data Availability

Data are contained within the article.

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
