# Peer review of "The Influence of Zinc Oxide Nanoparticles and Salt Stress on the Morphological and Some Biochemical Characteristics of Solanum lycopersicum L. Plants"

_plants, 2024, doi:10.3390/plants13101418_

Round 1
Reviewer 1 Report
Comments and Suggestions for Authors
I commend the authors for the analysis methods used, the analysis of the experimental data, the experimental concept and the topic addressed.
I propose the authors to continue and expand the research considering the complexity and importance of the field, as they mentioned in the conclusions of the study. Considering the socio-economic importance of the field, the studies should be directed, financed and carried out in the future and validated in experimental field and different pedo-climatic conditions.
However, special attention and new studies should be given regarding the possible phytotoxic effects and on food safety and security through the application of nanofertilizers, a fact that still limited their inclusion in the existing, legislated and approved categories of fertilizing products.
Author Response
Responses to Reviewer 1
Comments and Suggestions for Authors
I commend the authors for the analysis methods used, the analysis of the experimental data, the experimental concept and the topic addressed. I propose the authors to continue and expand the research considering the complexity and importance of the field, as they mentioned in the conclusions of the study. Considering the socio-economic importance of the field, the studies should be directed, financed and carried out in the future and validated in experimental field and different pedo-climatic conditions. However, special attention and new studies should be given regarding the possible phytotoxic effects and on food safety and security through the application of nanofertilizers, a fact that still limited their inclusion in the existing, legislated and approved categories of fertilizing products.
Thanks so much for your comments. We appreciate your contribution to improving our article. We are planning to perform the second phase of our study. We aim to continue and examine the studies using the chemically synthesized ZnO-NPs in the field on the same cultivar under different conditions, considering the complexity and importance of the field and the commerciality. Some recent articles do not recommend using the synthesized nanoparticles generally, trying to prove the phytotoxicity of those synthesized compounds, we were aware of that before designing our experiment and spent a considerable time collecting the previous published data that were discussing the assumed issue, analyzing their results, and finally, we reached the final conceptualization to begin our investigations. Our answer does not contradict that we were aware that we would have to do some further studies to examine the phytotoxic effects of the applied nanoparticles, we knew from the beginning that somehow, we would need to strengthen our obtained data, especially since the results were positively affecting some growth and photosynthetic attributes and showing a positive influence of the synthesized nanoparticles on reducing the stress markers in the salt-stressed plants. However, the expected toxic effects of nano-fertilizers on the food chain and, subsequently, on human health, especially under higher doses, still require more studies, which should use different crops and different kinds of nano-fertilizers. Using the zinc oxide nanoparticles was also a determining factor in our plans, zinc is a micro-crucial element that is needed in small amounts for the plants' development, and is a must for some enzymes such as; superoxide dismutase and catalase. In our study, we applied very small dosages of ZnO-NPs; 75 and 150 mg/L, they were foliar sprayed three times within the growing season, without adding them to the soil or accumulating over the fruits. Furthermore, we also are currently analyzing the transcriptome out of the leaves to get an extensive look at the subcellular alterations that may occurred to the plant leaves themselves.
Many thanks again.

Reviewer 2 Report
Comments and Suggestions for Authors
The manuscript is devoted to a rather important problem of our time, namely, increasing the productivity of soils containing excessive salt concentrations. The authors propose to solve the problem by adding nano-sized zinc oxide. The positive results in the manuscript are presented clearly and understandably, although in some places they are rather lengthy, with unnecessary details. There are currently a number of points in the manuscript that require improvement.
1. It seems to me that the introduction to the manuscript is rather contrived and drawn out. I am not aware of industrial models for the isolation of carotenoids or any other secondary metabolites from tomato fruits. This will probably never happen, since tomato fruits are tasty and in demand on their own, but there are a lot of other inedible plant materials rich in secondary metabolites. Probably the introduction needs to be rewritten, otherwise it turns out that the authors are solving a non-existent problem.
2. The second part of the introduction to the manuscript is devoted to salinization and is written clearly. Next, the authors continue the story about zinc oxide nanoparticles and zinc oxide in general. Zinc oxide is supposed to affect photosynthesis, etc. Yes, it is known that zinc oxide changes the expression of a number of genes in plants, but very often the positive effects of zinc oxide are associated with the effect of zinc oxide on the microbiome or pathogenic microflora. I recommend that authors take this aspect into account; to facilitate the authors’ work, I recommend the article (10.3367/UFNe.2023.09.039577).
3. Figure 5 raises many questions... Data obtained on a Malvern Instruments Zetasizer Nano-ZS. I have experience working with it, the graph looks more like a drawing than one obtained on the device. The autocorrelation function and polydispersity index are not shown. Frankly speaking, I doubt the possibility of obtaining an autocorrelation function for a graph with such creases... Also, the Malvern Instruments Zetasizer Nano-ZS does not allow you to obtain a size accurate to picometers, as the authors presented. I propose to delete this graph, and simply indicate the size of nanoparticles in the text of the manuscript, with an accuracy of nanometers. Write that the distribution is monomodal with a half-width of, for example, 20 nm. Unfortunately, the pictures obtained on the microscope also confirm my opinion.
4. The electrokinetic potential of the Malvern Nano-ZS can only measure down to tenths of a millivolt. Please take this into account and correct it.
5. Dates in tables are constantly presented in different formats, for example 5/7/2023 or 19-7-2023. It is important? Does this mean something? If not, please make everything the same!
6. Measuring the height of plants deserves special mention. The methods stated that plant lengths were measured to the nearest centimeter. Moreover, in the tables inside the manuscript, for example tables 4 and 5, the length is measured with an accuracy of hundredths of a centimeter. Obviously, such accuracy is not achievable with the help of a ruler or gauge... I suggest that the authors of the math game leave it to the mathematicians and correct the values in the tables themselves.
7. Why do the authors show Figure 9? What new things can he show us? I suggest transferring the drawing to the appendix.
8. Chromatographic standards and chromatograms are not given in the texts of the manuscript unless the authors have developed some new approach. Obviously, this manuscript does not present any new approach. The standard method is used. I suggest that the authors transfer all chromatograms to the appendix (appendix).
9. I will analyze the problems with measurements using the example of Table 8. The concentration of hydrogen peroxide is measured in nM or µM, since hydrogen peroxide is a molecule with a specific and known structure. If we recalculate the concentration of H2O2 in moles, then the concentration in T1 = 2 nM. SUCH low values are not achievable for living things. In principle, I would not believe in such values even in especially pure water... MDA (mmols/mL) from 100 to 1000 units, a concentration of 100 M is unattainable for MDA, simply mathematically not achievable. 100 M MDA takes up a volume of much more than 1 liter!!! Some may think that this is fraudulent... Authors should put all units of measurement in the manuscript in order.
Author Response
Responses to Reviewer 2
Comments and Suggestions for Authors
The manuscript is devoted to a rather important problem of our time, namely, increasing the productivity of soils containing excessive salt concentrations. The authors propose to solve the problem by adding nano-sized zinc oxide. The positive results in the manuscript are presented clearly and understandably, although in some places they are rather lengthy, with unnecessary details. There are currently a number of points in the manuscript that require improvement.
Thanks so much for your comments. We appreciate your contribution to improving our article. You will find the responses after every single comment. The required corrections were also included and highlighted in the manuscript.
1- It seems to me that the introduction to the manuscript is rather contrived and drawn out. I am not aware of industrial models for the isolation of carotenoids or any other secondary metabolites from tomato fruits. This will probably never happen, since tomato fruits are tasty and in demand on their own, but there are a lot of other inedible plant materials rich in secondary metabolites. Probably the introduction needs to be rewritten, otherwise it turns out that the authors are solving a non-existent problem.
We attempted to hint at the crucial role of the tomato plant and its abundance of beneficial compounds, highlighting the seriousness of the salinity stress problem's threat to the commercial plant's productivity. Thanks a lot for your comment. We carefully revised and rewrote the introduction to make it more concise and specific to the study's aim. We removed the secondary metabolites section to avoid unnecessary explanation and substituted it with valuable insights from the recommended article, which detailed the impact of metal and metal oxide nanoparticles on pathogens and the microbiome.
2- The second part of the introduction to the manuscript is devoted to salinization and is written clearly. Next, the authors continue the story about zinc oxide nanoparticles and zinc oxide in general. Zinc oxide is supposed to affect photosynthesis, etc. Yes, it is known that zinc oxide changes the expression of a number of genes in plants, but very often the positive effects of zinc oxide are associated with the effect of zinc oxide on the microbiome or pathogenic microflora. I recommend that authors take this aspect into account; to facilitate the authors’ work, I recommend the article (10.3367/UFNe.2023.09.039577).
Thanks so much for mentioning such a useful article, it was so helpful. The article guided us toward an alternative perspective on the use of metal oxide nanoparticles, such as zinc oxide nanoparticles, in several industries, including the food sector. Applications in the food industry may involve the creation of novel biodegradable packaging materials that possess antibacterial qualities, as well as the development of protective coatings for work surfaces, specifically in the context of meat processing. It is one of our current works on using those chemically synthesized zinc oxide nanoparticles as antimicrobial agents and also comparing the effect of those particles and the effect of the extracts from the leaves that were sprayed with them. The above-mentioned article was really helpful, and the discussed part was supported by some of its conclusions.
3- Figure 5 raises many questions... Data obtained on a Malvern Instruments Zetasizer Nano-ZS. I have experience working with it, the graph looks more like a drawing than one obtained on the device. The autocorrelation function and polydispersity index are not shown. Frankly speaking, I doubt the possibility of obtaining an autocorrelation function for a graph with such creases... Also, the Malvern Instruments Zetasizer Nano-ZS does not allow you to obtain a size accurate to picometers, as the authors presented. I propose to delete this graph, and simply indicate the size of nanoparticles in the text of the manuscript, with an accuracy of nanometers. Write that the distribution is monomodal with a half-width of, for example, 20 nm. Unfortunately, the pictures obtained on the microscope also confirm my opinion.
Thanks so much for your comment. We have also used the original charts from the Malvern-Nano size apparatus in the past, whether we were measuring the size (d.nm) or the zeta potential (mV). Since we only had a printed hard copy of the analysis during the current study, we tried to use some other software like Origin or Excel to draw the results of the size and charge characterization. We have taken note of your feedback, removed the fifth figure as suggested, and incorporated the size result into the text.
- The electrokinetic potential of the Malvern Nano-ZS can only measure down to tenths of a millivolt. Please take this into account and correct it.
Thanks a lot for your comment. The figure was corrected.
- Dates in tables are constantly presented in different formats, for example 5/7/2023 or 19-7-2023. It is important? Does this mean something? If not, please make everything the same!
No, it does not mean something specific. Thanks for the notice. Presenting the dates in all parts of the manuscript was unified.
- Measuring the height of plants deserves special mention. The methods stated that plant lengths were measured to the nearest centimeter. Moreover, in the tables inside the manuscript, for example tables 4 and 5, the length is measured with an accuracy of hundredths of a centimeter. Obviously, such accuracy is not achievable with the help of a ruler or gauge... I suggest that the authors of the math game leave it to the mathematicians and correct the values in the tables themselves.
Your comments and suggestions are appreciated. We want to clarify that the raw data before the analysis were collected from the plants according to the nearest centimeters, for example, the data of the plant height of the plants in the first treatment were -in every stage of collecting- as follows:
|
T1 |
Plant height 1 |
Plant height 2 |
Plant height 3 |
Plant height 4 |
|
28.2 |
45 |
60 |
62 |
|
|
28 |
49 |
54 |
55 |
|
|
24 |
50 |
62 |
64 |
|
|
19 |
46 |
55 |
58 |
For the stem width, we used the caliber, it provided us with the millimeter values, but we also collected the results according to the nearest centimeters as follows:
|
T1 |
Stem width 1 |
Stem width 2 |
Stem width 3 |
Stem width 4 |
|
0.5 |
0.9 |
1 |
1.2 |
|
|
0.5 |
0.8 |
1 |
1.1 |
|
|
0.4 |
0.6 |
0.9 |
1.1 |
|
|
0.5 |
0.6 |
0.9 |
1.2 |
Since we had 4 replicates in each treatment, we collected 4 readings from each treatment at every stage. So, it was much easier for us to include the means of the readings without analyzing the values to find the standard errors and significance. We just wanted to unify the statistical analytical method for all the readings in different tables. We recommend including the following sentence "Each value represents the mean±SE to the approximate centimeters" in the caption below the tables 4 and 5. We really appreciate your suggestion, and we do not mind repeating the whole statistical analysis for the two mentioned tables as you suggested, we are recommending, but we are still open to recalculate the values if you see that it is a must.
- Why do the authors show Figure 9? What new things can he show us? I suggest transferring the drawing to the appendix.
Thanks. The figure was transferred to Appendix A.
- Chromatographic standards and chromatograms are not given in the texts of the manuscript unless the authors have developed some new approach. Obviously, this manuscript does not present any new approach. The standard method is used. I suggest that the authors transfer all chromatograms to the appendix (appendix).
Thanks. The chromatograms were transferred to Appendix A.
- I will analyze the problems with measurements using the example of Table 8. The concentration of hydrogen peroxide is measured in nM or µM, since hydrogen peroxide is a molecule with a specific and known structure. If we recalculate the concentration of H2O2 in moles, then the concentration in T1 = 2 nM. SUCH low values are not achievable for living things. In principle, I would not believe in such values even in especially pure water... MDA (mmols/mL) from 100 to 1000 units, a concentration of 100 M is unattainable for MDA, simply mathematically not achievable. 100 M MDA takes up a volume of much more than 1 liter!!! Some may think that this is fraudulent... Authors should put all units of measurement in the manuscript in order.
Thanks so much for your comment. First of all, we understand your comments well. And we are so grateful for considering our article and providing us with such valuable points that non-intentionally happened. We hope you find our coming detailed answer acceptable. Again, we appreciate your precious time in finding that defect in calculating the above-mentioned molecules; H2O2 and malondialdehyde. We also hope that you find our long answer not inconvenient.
Regarding the malondialdehyde, the first flaw was presenting the results in mmol/mL, while our used protocol was certifying mmols MDA g-1FW as a unit of measurement. So, the units were revised and modified to be mmols MDA g-1FW. The second flaw was that we tried to applied the Lambert-Beer extinction coefficient as εΜ= 1.55 *10^6 mM-1cm-1, following the next equation:
(ΔA 532nm/sample – ΔA 600nm/ sample)
Extinction coefficient (εΜ) * Path light (L)
Since 1.55 *10^5 M^−1cm^−1 equals 155*10^6 mM^−1cm^−1, we calculated the results as mmole/mL using the wrong equation number, the calculations in the spreadsheet were done using the coefficient 1.55*10^6 mM^−1cm^−1 instead of 155*10^6 mM^−1cm^−1. So, the results were much greater than they were supposed to be. The values of MDA in Table 8 were corrected. The original absorbances of the samples at 532 and 600 nm are the same. The notations are the same and the LSD (0.01 and 0.05) were corrected according to the corrected values.
On the other side, H2O2 is usually expressed as microgram per gram FW or micromole per gram FW). [μmol g^−1 FW or μg g^−1 FW]. In our study, it was determined based on μg g^−1 FW. The flaw also was in the calculation step, when we extracted H2O2 in TCA, we followed the extraction by transferring 1 mL, the values in Table 8 were expressed as μg/g, while they represented the amount in 1 mL that was transferred, by repeating the calculations to measure the compound in 1 g, not in 0.1 g, we got the actual values and they are modified in the table, and the modification involved the new statistical analysis to get the new errors of the values, it was also a mathematical flaw.
We would like to point out that the previous two notes were very important to us. They prompted us to reconsider the rest of the values ​​in the five tables, especially Table 8, restore the original readings, and retrieve the protocols that were used in the measurements. Honestly, your comments were very professional and precise for us, and they led us toward a wonderful improvement to our manuscript.
We just wanted to make it clear that it was an expanded study, and many analyses have been conducted through the authors, of course, that unintentional flaw had to be noticed earlier, so we are so grateful for your honest comments and suggestions.
We also like to present our procedures in detail, it is just to demonstrate how we estimated those molecules, and we are available and welcome any further suggestions to improve our article, it will be our pleasure. Kindly, if you like to take look on the following protocols that were used to determine the above-discussed molecules, it will be highly appreciated.
- Malondialdehyde (MDA) procedures
Reagents
- 1% (w/v) TCA (0.5 g TCA in 500 mL).
- 5% ΤΒΑ diluted in 20 % (w/v) TCA. To prepare 20% TCA; add 30 g of TCA in 150 mL d.w, mix well, then add 0.75 g of TBA to the previous solution).
Steps
- Homogenize 0.5 g (500 mg) of leaf tissue by adding 5 mL 0.1 % (w/v) TCA.
- Centrifuge the homogenate for 10 min (13000 rpm at 4℃).
- Falcon tubes were used, and small holes were created in the cap using a syringe needle to prevent the Falcons from bursting due to pressure from the heat.
- Collect supernatant and mix 0.8 mL (800 µL) of supernatant with 2 mL of 0.5% ΤΒΑ diluted in 20 % TCA. Incubate in a water bath at 80℃ for 25 min.
- End reaction by incubating on ice for 5 min. In case the solution is not clear, centrifuge for a further 5 min (13000 rpm at 4℃).
- Measure the absorbance of the pink color at 532 (the maximum absorbance of MDA) and 600 nm (which represents the turbidity). OD600 values are subtracted from the MDA-TBA complex values at 532 nm and MDA concentration is calculated using the Lambert-Beer law with an extinction coefficient εΜ= 155 * 10^6 mM-1cm-1. Results are presented as mmols MDA/mL.
2- Hydrogen peroxide (H2O2) procedures
Reagents
- 1% (w/v) TCA (0.5 g TCA in 500 mL).
- phosphate buffer (100 mM, pH 7.2); 0.6 g of NaH2PO4 and 0.71 g of Na2HPO4 in 70 mL of d.w, adjust the pH, and then complete the volume to 100 mL with d.w.
- KI (1 M); 16.6 g of KI in 100 mL d.w.
Steps
- Fresh leaves weighing about 0.5 g (500 mg) were directly homogenized with 5 mL of Trichloroacetic acid (TCA) (0.1% (w/v) solution.
- Centrifuge for 15 minutes at 12000 rpm at 4℃.
- 1 mL KI (1 M) and 1 mL phosphate buffer (100 mM, pH 7.2) were added to 1 mL of the supernatant obtained from step 1.
- At the same time, for every sample, a control was prepared with H2O instead of KI for the tissue coloration background.
- Vortex of this mixture and left to incubate at room temperature (20°C - 22°C) for 20 min.
- The samples’ absorbance of the straw-formed color was measured at 390 nm.
- A calibration curve obtained with H2O2 (μg/mL) standard solutions prepared in 0.1% TCA was used for quantification in a stock of 100 μg/mL. The H2O2 concentration can be determined using a standard concentration curve and calculated on a fresh weight basis (usually expressed as microgram per gram FW or micromole per gram FW). [μmol g^−1 FW or μg g^−1 FW].

Reviewer 3 Report
Comments and Suggestions for Authors
Dear Authors
I read with great interest the manuscript by Mostafa Ahmed et al, “Understanding the Effect of Salinity on Solanum lycopersicum L. Plants Cultivated in a Greenhouse and Studying the Altering Impact of Zinc Oxide Nanoparticles on Different Biochemical and Morphological Characteristics of Their Leaves.” In general, the research topic is relevant; the authors obtained a lot of experimental results, which are certainly interesting. The data obtained suggest that the use of ZnO-NPs reduces the negative effects of salt stress on tomato plants. It seems to me that the resulting zinc oxide nanoparticles are quite well characterized.
At the same time, the presentation of the material requires serious adjustments. The volume of the manuscript is unreasonably large. The text and even subheadings are overloaded with unnecessary words that need to be removed. As an example, I will give the title of the manuscript (see above). I think that it is necessary to remove words that lack informational load. As one of the options for discussion, I could suggest the following: “The influence of zinc oxide nanoparticles and salt stress on the morphological and some biochemical characteristics of Solanum lycopersicum L.” plants.
The manuscript is difficult to understand. Thus, the annotation uses a lot of abbreviations that the reader is not familiar with, so it would be better to give full names in this section. A lot of attention is paid to justifying the fact that in order to provide food for the population, it is necessary to increase the resistance of plants, including through the use of nanoparticles.
Unfortunately, the authors did not formulate a working hypothesis, which would have made the material easier to understand and would have made the work more coherent. The purpose of the work is not formulated very well. The authors planned to study photosynthetic activity in this work, although they did not do this. They measured chlorophyll content, but this is not called photosynthetic activity. However, in the conclusion the authors again state that they were studying photosynthetic activity.
The Materials and Methods section is written in too much detail. Standard methods that have been used for decades do not require a detailed description, as is done in a laboratory journal. The authors’ desire to describe in detail different parts of the work, for example, the section “Determining the amount of air-dried soil to place in each pot,” is respectable. Obviously, to make the experiment easier to understand, the authors introduced a very useful section called Experimental profile; unfortunately, it does not add any clarity. I think it would be very helpful to present a flowchart for the experiments rather than just listing dates.
I would like to note that almost nowhere is the manufacturer of the devices used in the work indicated; the centrifugation speed is usually expressed in g, and not in rpm.
Particular problems are caused by presenting the content of certain components not in molar dimensions, not in the content of a particular compound or element per g (mg) of dry or fresh weight, but in ppm. This does not allow the reader to compare the results presented with literature data.
It is not clear why it is necessary to calculate the chlorophyll content per leaf area, and it is even more unclear why, under fairly strong salinity, there is no inhibition of the leaf surface of tomato plants.
In addition, it is not described anywhere on what basis the authors identified 4 stages in the ontogeny of tomatoes
It is not clear why a concentration of sodium chloride of 150 mM and concentrations of zinc oxide nanoparticles of 75 and 150 mg/l were chosen for the experiments. It is also not indicated how the required amount of salt was added - one time or in several doses.
It is imperative to reduce the number of chromatograms presented in the main text. You can cite 1-2, and put all the rest in the appendix to the article. I mentioned only a few comments. It is extremely difficult to list everything, and it is not necessary.
One of the serious shortcomings of this study is that the authors did not attempt to assess the water status of plants. This is very important, since salt causes osmotic stress in plants, resulting in the development of water deficiency. This could be achieved by assessing tissue water content, relative water content and, ideally, osmotic or water potential.
However, even without data on water status, I believe that the results obtained are more than sufficient for publication. The authors should try to present it more concisely and clearly. I believe that the authors will undergo significant editing of the submitted manuscript. In its present form it cannot be recommended for publication.
Kind regards
Author Response
Responses to Reviewer 3
Comments and Suggestions for Authors
Dear Authors
- I read with great interest the manuscript by Mostafa Ahmed et al, “Understanding the Effect of Salinity on Solanum lycopersicum L. Plants Cultivated in a Greenhouse and Studying the Altering Impact of Zinc Oxide Nanoparticles on Different Biochemical and Morphological Characteristics of Their Leaves.” In general, the research topic is relevant; the authors obtained a lot of experimental results, which are certainly interesting. The data obtained suggest that the use of ZnO-NPs reduces the negative effects of salt stress on tomato plants. It seems to me that the resulting zinc oxide nanoparticles are quite well characterized.
Thanks so much for your consideration of our manuscript and for reading it interestingly. We appreciate that, and we are excited to follow your comments, suggestions, and required corrections. We will take it seriously as a chance to improve our work and promise to do our best regarding the focused points.
- At the same time, the presentation of the material requires serious adjustments. The volume of the manuscript is unreasonably large. The text and even subheadings are overloaded with unnecessary words that need to be removed. As an example, I will give the title of the manuscript (see above). I think that it is necessary to remove words that lack informational load. As one of the options for discussion, I could suggest the following: “The influence of zinc oxide nanoparticles and salt stress on the morphological and some biochemical characteristics of Solanum lycopersicum L.” plants.
Thanks a lot for your comment and suggestion. The manuscript was carefully revised to avoid unnecessary words. We appreciate the suggested title, the old one was replaced with it. We also merged many subheadings within the manuscript in different sections, especially; results and materials and methods.
- The manuscript is difficult to understand.Thus, the annotation uses a lot of abbreviations that the reader is not familiar with, so it would be better to give full names in this section. A lot of attention is paid to justifying the fact that in order to provide food for the population, it is necessary to increase the resistance of plants, including through the use of nanoparticles.
Thanks so much for your suggestion. We revised the whole manuscript to make sure that all the abbreviations were mentioned the first time in their full definitions, and we also included an abbreviation list before the introduction section. Of course, it is crucial, since the discussed topic in the manuscript is being widely used to provide food for the population.
- Unfortunately, the authors did not formulate a working hypothesis, which would have made the material easier to understand and would have made the work more coherent. The purpose of the work is not formulated very well. The authors planned to study photosynthetic activity in this work, although they did not do this. They measured chlorophyll content, but this is not called photosynthetic activity. However, in the conclusion the authors again state that they were studying photosynthetic activity.
Thanks a lot for that critical comment. We aimed to estimate the chlorophyll content as a factor of the photosynthetic attributes, and an indicator of the photosynthetic activity. We did not mean that our purpose of the study is to go for a wide investigation on the photosynthetic pigments, carotenoids, Chl a and Chl b, Pn, Pr, stomatal closure and internal carbon dioxide, etc. It was a non-intentional conclusive point from our side. The aim of the study was changed in the abstract, introduction, and conclusions sections. Any other sections showing the flaws of the photosynthetic activity; were carefully revised and modified to fit the findings of our study.
- The Materials and Methods sectionis written in too much detail. Standard methods that have been used for decades do not require a detailed description, as is done in a laboratory journal. The authors’ desire to describe in detail different parts of the work, for example, the section “Determining the amount of air-dried soil to place in each pot,” is respectable. Obviously, to make the experiment easier to understand, the authors introduced a very useful section called Experimental profile; unfortunately, it does not add any clarity. I think it would be very helpful to present a flowchart for the experiments rather than just listing dates.
We appreciate your suggestions, many thanks for the comment. The materials and methods section was summarized to avoid some unnecessary details. Also, the part of the "Experimental profile" was removed. We tried to merge some subheadings in the above-mentioned section to make it more concise.
- I would like to note that almost nowhere is the manufacturer of the devices used in the work indicated; the centrifugation speed is usually expressed in g, and not in rpm.
Many thanks for the notice, the centrifugation speed was converted to g force, and the manufacturers were added to the devices used in different analyses.
- Particular problems are caused by presenting the content of certain components not in molar dimensions, not in the content of a particular compound or element per g (mg) of dry or fresh weight, but in ppm. This does not allow the reader to compare the results presented with literature data.
Only the micro and macro elements were presented in ppm. We modified the units of the values to be expressed as microgram/g, as 1 microgram/1gram= 1 ppm. All the required changes in the text were taken into account in the revision. It was important to unify the units. Thanks a lot for the notice.
- It is not clear why it is necessary to calculate the chlorophyll content per leaf area, and it is even more unclear why, under fairly strong salinity, there is no inhibition of the leaf surface of tomato plants.
Thanks so much for your comment. The readings of chlorophyll were recorded in SPAD units. SPAD values were recorded according to the biggest leaf on all its leaflets. We did calculate the Chl content per leaf area because we just tried to reduce the error by picking many recordings from the same leaf, and the biggest newly formed leaf had several leaflets. We used to use the SPAD apparatus 10 times on the same leaf, then to get the average. So, we almost got 40 recordings per treatment.
Regarding the leaf area, indeed, there was an obvious inhibition in the leaf area, you can take a look at Figures A1-1 and A1-2 in the Appendix. The main reason for the non-significant of the values was the high standard errors that were calculated. We stated -in the manuscript before reviewing- that there was an increasing manner in the leaf area within the different stages, and there was an obvious increase in the leaf area of the non-stressed treatments (T1-T3) during the study compared to the stressed treatments (T4-T6), but statistically, it was not approved, as the replicates from each treatment had a value that might affect the values of the standard error. After reviewing, we repeated the whole statistical analysis to reduce the standard error. Of course, the raw data are the same, we just excluded the replicates that were extremely different, they were responsible for the high errors in the values, which caused unusual non-significance. We hope when you revise the corrected values in the table demonstrating the leaf area, to take the values -before correction- into account, and also take the above-mentioned statement into account, it is just a mathematical correction, no doubt that the values are still the outcomings from the same raw data. The manner of increasing and reducing the leaf is the same in the non-stressed treatments and stressed treatments, respectively. We confirm that we preferred to present the values with non-significance in the manuscript before reviewing because we used a manual method to assess the leaf area, it was a belief to leave the values as they are without excluding the extreme values when we did the statistical analysis. Please find the data on the leaf area and its analysis at the end of this file (after the comments).
- In addition, it is not described anywhere on what basis the authors identified 4 stages in the ontogeny of tomatoes
Thanks a lot for your notice. In the subheading called experimental profile in the materials and methods section, we stated the specific dates of different treatments, beginning with the sowing and transplanting to the bigger pots. We thought that was enough to show that we chose those different stages based on the vegetative and generative phases of the tomatoes. But, it was not enough, thanks a lot for the notice. We included that critical point in subheading 4.5 Determination of chlorophyll and growth attributes. Your comment was crucial in including the reason for spraying the chemically synthesized ZnO-NPs three times on specific dates. We also included that point in the subheading of the experimental design.
- It is not clear why a concentration of sodium chloride of 150 mM and concentrations of zinc oxide nanoparticles of 75 and 150 mg/l were chosen for the experiments. It is also not indicated how the required amount of salt was added - one time or in several doses.
Thanks for the comment. Since tomatoes are a moderately sensitive plant to salinity, so we chose the concentration (15 ds/m) that causes severe salinity stress. Several studies confirmed that the threshold of tolerance of tomatoes to salt in irrigation water is 1.7-2.5 ds/m and the slope of yield decrease of increasing the salinity by 1 ds/m is 9.9 (% per dS/m) (https://doi.org/10.3390/horticulturae3020030 and https://doi:10.1126/science.1183700). So, we chose the concentration 150 mM (15 ds/m), which was also used in some other studies (https://doi.org/10.1016/j.plaphy.2021.02.002). We mentioned in the design subheading that NaCl (150 mM) stress solution was applied in the soil on the 10th day after the transplanting (40 DAS) to provide salt stress. To make it clear, kindly follow the following table, to get the concept of the basis of irrigation process.
We were irrigating the stressed and non-stressed treatments constantly, using the TDR readings. And, to unify soil electric conductivity (EC), EC water drainage was monitored daily for all treatments and it was maintained equal to the EC irrigating solution. That was done by allowing enough drainage from the root zone by the use of a corresponding saline solution until equilibrium between the EC of water drainage and irrigating solution. In that way, the EC root zone stabilized at the specified set point during the experiment. Consequently, the amount of water added at each time ranged between 1 to 1.5 Liters. It was adjusted based on the EC water drainage obtained at each time.
In the case of the concentrations of ZnO-NPs that were used (75 and 150 mg/L). It was also according to the collected data from the previous literature, like the above-mentioned article (https://doi.org/10.1016/j.plaphy.2021.02.002). The authors used 3 different concentrations, and we made them 2, the lowest was not used in their study (75 mg /L), and it duplicated one (150 mg/L), which its effect was already examined in their study.
- It is imperative to reduce the number of chromatograms presented in the main text. You can cite 1-2, and put all the rest in the appendix to the article. I mentioned only a few comments. It is extremely difficult to list everything, and it is not necessary.
We appreciate your concern and consideration of our article. We found your comments helpful in improving the manuscript and enlightened us with many points. The chromatograms and figures 9-1 and 9-2 were transferred to the appendix.
- One of the serious shortcomings of this study is that the authors did not attempt to assess the water status of plants. This is very important, since salt causes osmotic stress in plants, resulting in the development of water deficiency. This could be achieved by assessing tissue water content, relative water content and, ideally, osmotic or water potential.
Thanks a lot for the suggestion. It would be very fruitful to assess the water status. Since the study focused on determining the biochemical markers and the growth attributes in the presence or absence of salt/ZnO-NPs, we did not perform some useful analyses that obviously would improve the promising outcomes of the study. But, we tried to use that phase to investigate Biochemistry and a few morphological factors, besides the synthesis and characterization of the nanoparticles. We plan to perform the second phase of our study by examining some other parameters There are also some other analyses ongoing; sequencing, and transcriptomic analysis. So, we aimed to focus on that field. Since we no longer have the fresh cuts from the tomato leaves, we promise to take the mentioned analyses into account in our upcoming studies.
- However, even without data on water status, I believe that the results obtained are more than sufficient for publication. The authors should try to present it more concisely and clearly. I believe that the authors will undergo significant editing of the submitted manuscript. In its present form it cannot be recommended for publication. Kind regards
Thanks so much for your supporting comments. We honestly are grateful and hope the responses and corrections already improved the manuscript to be considered for further steps.
|
Raw data of the leaf are recordings (before reviewing) (With the extreme values that cause high standard errors which led to lack of significance) |
||||
|
T1 |
Stage 1 |
Stage 2 |
Stage 3 |
Stage 4 |
|
Leaf area |
Leaf area |
Leaf area |
Leaf area |
|
|
160 |
345 |
610.5 |
613 |
|
|
123.5 |
432.3 |
511.5 |
612 |
|
|
115.625 |
464 |
510 |
580.2 |
|
|
107.25 |
194.25 |
516 |
577.5 |
|
|
T2 |
Stage 1 |
Stage 2 |
Stage 3 |
Stage 4 |
|
Leaf area |
Leaf area |
Leaf area |
Leaf area |
|
|
184 |
433.95 |
485.75 |
632.7 |
|
|
181.25 |
435.5 |
478.5 |
643.5 |
|
|
121.5 |
485.75 |
495 |
632.7 |
|
|
147.25 |
377.625 |
478.5 |
643.5 |
|
|
T3 |
Stage 1 |
Stage 2 |
Stage 3 |
Stage 4 |
|
Leaf area |
Leaf area |
Leaf area |
Leaf area |
|
|
137.75 |
419.75 |
577.5 |
615 |
|
|
135 |
409.5 |
610.5 |
649.3 |
|
|
112 |
581 |
638.75 |
647.5 |
|
|
110.5 |
542.5 |
610.5 |
613.3 |
|
|
T4 |
Stage 1 |
Stage 2 |
Stage 3 |
Stage 4 |
|
Leaf area |
Leaf area |
Leaf area |
Leaf area |
|
|
173.25 |
378 |
412.5 |
493 |
|
|
136.5 |
325 |
372 |
412.5 |
|
|
216 |
448 |
527 |
527 |
|
|
80.75 |
286 |
372 |
493 |
|
|
T5 |
Stage 1 |
Stage 2 |
Stage 3 |
Stage 4 |
|
Leaf area |
Leaf area |
Leaf area |
Leaf area |
|
|
133.25 |
357.75 |
435 |
452.6 |
|
|
130 |
364 |
449.5 |
569 |
|
|
181.5 |
440 |
520 |
561 |
|
|
155 |
398.75 |
435 |
442 |
|
|
T6 |
Stage 1 |
Stage 2 |
Stage 3 |
Stage 4 |
|
Leaf area |
Leaf area |
Leaf area |
Leaf area |
|
|
120 |
450 |
511.5 |
519 |
|
|
152.25 |
420.375 |
448 |
450 |
|
|
92 |
364 |
435 |
665 |
|
|
106.375 |
511.5 |
527 |
670 |
|
|
Raw data of the leaf are recordings (before reviewing) (With the extreme values that cause high standard errors which led to lack of significance) |
||||
|
T1 |
Stage 1 |
Stage 2 |
Stage 3 |
Stage 4 |
|
Leaf area |
Leaf area |
Leaf area |
Leaf area |
|
|
123.5 |
432.3 |
511.5 |
612 |
|
|
115.625 |
464 |
510 |
580.2 |
|
|
T2 |
Stage 1 |
Stage 2 |
Stage 3 |
Stage 4 |
|
Leaf area |
Leaf area |
Leaf area |
Leaf area |
|
|
184 |
433.95 |
485.75 |
632.7 |
|
|
181.25 |
435.5 |
478.5 |
643.5 |
|
|
T3 |
Stage 1 |
Stage 2 |
Stage 3 |
Stage 4 |
|
Leaf area |
Leaf area |
Leaf area |
Leaf area |
|
|
137.75 |
419.75 |
577.5 |
615 |
|
|
135 |
409.5 |
610.5 |
649.3 |
|
|
T4 |
Stage 1 |
Stage 2 |
Stage 3 |
Stage 4 |
|
Leaf area |
Leaf area |
Leaf area |
Leaf area |
|
|
173.25 |
378 |
412.5 |
493 |
|
|
136.5 |
325 |
372 |
412.5 |
|
|
T5 |
Stage 1 |
Stage 2 |
Stage 3 |
Stage 4 |
|
Leaf area |
Leaf area |
Leaf area |
Leaf area |
|
|
133.25 |
357.75 |
435 |
452.6 |
|
|
130 |
364 |
449.5 |
569 |
|
|
T6 |
Stage 1 |
Stage 2 |
Stage 3 |
Stage 4 |
|
Leaf area |
Leaf area |
Leaf area |
Leaf area |
|
|
120 |
450 |
511.5 |
519 |
|
|
152.25 |
420.375 |
448 |
450 |
|
|
Treatment means |
|
|
S.No |
Average |
|
Treatment 1 |
119.563 |
|
Treatment 2 |
182.625 |
|
Treatment 3 |
136.375 |
|
Treatment 4 |
154.875 |
|
Treatment 5 |
131.625 |
|
Treatment 6 |
136.125 |
Analysis after correction, Stage 1 for the different 6 treatments
|
|
|||||
|
Source of variation |
Degrees of freedom |
Sum of squares |
Mean sum of squares |
F cal |
F prob |
|
Treatments |
5 |
4958.652 |
991.730 |
4.802 |
0.041 |
|
Error |
6 |
1239.164 |
206.527 |
- |
- |
|
Total |
11 |
- |
- |
- |
- |
Coefficient of Variation = 10.012
|
Treatments found Significant at 5% level of Significance CD(0.05)= 35.166
|
Analysis after correction, Stage 2 for the different 6 treatments
|
Treatment means |
|
|
S.No |
Average |
|
Treatment 1 |
448.150 |
|
Treatment 2 |
434.725 |
|
Treatment 3 |
414.625 |
|
Treatment 4 |
351.500 |
|
Treatment 5 |
360.875 |
|
Treatment 6 |
435.188 |
|
Anova Table |
|||||
|
Source of variation |
Degrees of freedom |
Sum of squares |
Mean sum of squares |
F cal |
F prob |
|
Treatments |
5 |
17041.753 |
3408.351 |
8.454 |
0.011 |
|
Error |
6 |
2419.029 |
403.172 |
- |
- |
|
Total |
11 |
- |
- |
- |
- |
Coefficient of Variation = 4.927
|
Treatments found Significant at 5% level of Significance CD(0.05)= 49.134
|
Analysis after correction, Stage 3 for the different 6 treatments
|
Treatment means |
|
|
S.No |
Average |
|
Treatment 1 |
510.750 |
|
Treatment 2 |
482.125 |
|
Treatment 3 |
594.000 |
|
Treatment 4 |
392.250 |
|
Treatment 5 |
442.250 |
|
Treatment 6 |
479.750 |
|
Anova Table |
|||||
|
Source of variation |
Degrees of freedom |
Sum of squares |
Mean sum of squares |
F cal |
F prob |
|
Treatments |
5 |
45993.776 |
9198.755 |
15.710 |
0.002 |
|
Error |
6 |
3513.281 |
585.547 |
- |
- |
|
Total |
11 |
- |
- |
- |
- |
Coefficient of Variation = 5.005
|
Treatments found Significant at 1% and 5% level of significance
|
Analysis after correction, Stage 4 for the different 6 treatments
|
Treatment means |
|
|
S.No |
Average |
|
Treatment 1 |
596.100 |
|
Treatment 2 |
638.100 |
|
Treatment 3 |
632.150 |
|
Treatment 4 |
452.750 |
|
Treatment 5 |
510.800 |
|
Treatment 6 |
484.500 |
|
Anova Table |
|||||
|
Source of variation |
Degrees of freedom |
Sum of squares |
Mean sum of squares |
F cal |
F prob |
|
Treatments |
5 |
63770.670 |
12754.134 |
5.649 |
0.029 |
|
Error |
6 |
13547.290 |
2257.882 |
- |
- |
|
Total |
11 |
- |
- |
- |
- |
Coefficient of Variation = 8.602
|
Treatments found Significant at 5% level of Significance CD(0.05)= 116.275
|
|
Treatment means |
|
|
S.No |
Average |
|
Treatment 1 |
126.594 |
|
Treatment 2 |
158.500 |
|
Treatment 3 |
123.813 |
|
Treatment 4 |
151.625 |
|
Treatment 5 |
124.938 |
|
Treatment 6 |
117.656 |
Analysis before corrections, Stage 1 for the different 6 treatments
|
Anova Table |
|||||
|
Source of variation |
Degrees of freedom |
Sum of squares |
Mean sum of squares |
F cal |
F prob |
|
Treatments |
5 |
5674.591 |
1134.918 |
0.819 |
0.552 |
|
Error |
18 |
24956.430 |
1386.468 |
- |
- |
|
Total |
23 |
- |
- |
- |
- |
Coefficient of Variation = 27.818
|
Treatments found to be Non Significant |
Analysis before corrections, Stage 2 for the different 6 treatments
|
Treatment means |
|
|
S.No |
Average |
|
Treatment 1 |
358.888 |
|
Treatment 2 |
435.519 |
|
Treatment 3 |
502.625 |
|
Treatment 4 |
359.250 |
|
Treatment 5 |
410.125 |
|
Treatment 6 |
440.344 |
|
Anova Table |
|||||
|
Source of variation |
Degrees of freedom |
Sum of squares |
Mean sum of squares |
F cal |
F prob |
|
Treatments |
5 |
59900.583 |
11980.117 |
1.634 |
0.202 |
|
Error |
18 |
131958.112 |
7331.006 |
- |
- |
|
Total |
23 |
- |
- |
- |
- |
Coefficient of Variation = 20.494
|
Treatments found to be Non Significant |
Analysis before corrections, Stage 3 for the different 6 treatments
|
Treatment means |
|
|
S.No |
Average |
|
Treatment 1 |
552.375 |
|
Treatment 2 |
560.113 |
|
Treatment 3 |
611.500 |
|
Treatment 4 |
476.125 |
|
Treatment 5 |
470.125 |
|
Treatment 6 |
514.875 |
|
Anova Table |
|||||
|
Source of variation |
Degrees of freedom |
Sum of squares |
Mean sum of squares |
F cal |
F prob |
|
Treatments |
5 |
59046.376 |
11809.275 |
2.263 |
0.092 |
|
Error |
18 |
93933.552 |
5218.531 |
- |
- |
|
Total |
23 |
- |
- |
- |
- |
Coefficient of Variation = 13.608
|
Treatments found to be Non Significant |
Analysis before corrections, Stage 4 for the different 6 treatments
|
Treatment means |
|
|
S.No |
Average |
|
Treatment 1 |
555.300 |
|
Treatment 2 |
560.113 |
|
Treatment 3 |
614.650 |
|
Treatment 4 |
451.125 |
|
Treatment 5 |
475.900 |
|
Treatment 6 |
520.250 |
|
Anova Table |
|||||
|
Source of variation |
Degrees of freedom |
Sum of squares |
Mean sum of squares |
F cal |
F prob |
|
Treatments |
5 |
71817.727 |
14363.545 |
2.741 |
0.052 |
|
Error |
18 |
94312.829 |
5239.602 |
- |
- |
|
Total |
23 |
- |
- |
- |
- |
Coefficient of Variation = 13.669
|
Treatments found to be Non Significant |

Reviewer 4 Report
Comments and Suggestions for Authors
Salinity is well known for reducing crop yield and quality, causing global economic losses. Zinc oxide nano-particles (ZnO-NPs) was reported to improve plant physiological and metabolic processes and abiotic stress resistance. This study examined how foliar application of chemically synthesized ZnO-NPs to the leaves affected biochemistry, morphology, photosynthetic activity, and phenolic compound synthesis with and without NaCl in tomato and produce a significant results. In general, this paper was well written and experimental design is good, results are significant and I have the following comments:
Major comments:
The authors should check the expression pattern and slat stressed related genes.
Minor comment:
1. For the macro and micro-elements, why not check the N and P contents?
2. The first letter of any key word should be capital;
3. Figures 11-16 show, show should be the passed tense showed, please check the others.
Comments on the Quality of English LanguageSalinity is well known for reducing crop yield and quality, causing global economic losses. Zinc oxide nano-particles (ZnO-NPs) was reported to improve plant physiological and metabolic processes and abiotic stress resistance. This study examined how foliar application of chemically synthesized ZnO-NPs to the leaves affected biochemistry, morphology, photosynthetic activity, and phenolic compound synthesis with and without NaCl in tomato and produce a significant results. In general, this paper was well written and experimental design is good, results are significant and I have the following comments:
Major comments:
The authors should check the expression pattern and slat stressed related genes.
Minor comment:
1. For the macro and micro-elements, why not check the N and P contents?
2. The first letter of any key word should be capital;
3. Figures 11-16 show, show should be the passed tense showed, please check the others.
Author Response
Responses to Reviewer 4
Comments and Suggestions for Authors
Salinity is well known for reducing crop yield and quality, causing global economic losses. Zinc oxide nano-particles (ZnO-NPs) was reported to improve plant physiological and metabolic processes and abiotic stress resistance. This study examined how foliar application of chemically synthesized ZnO-NPs to the leaves affected biochemistry, morphology, photosynthetic activity, and phenolic compound synthesis with and without NaCl in tomato and produce a significant results. In general, this paper was well written and experimental design is good, results are significant.
Thanks so much for your comments. We appreciate your full consideration and contribution to improving our article.
I have the following comments:
Major comments:
- The authors should check the expression pattern and slat stressed related genes.
Thanks a lot for your comment. We are planning to perform the second phase of our study. We aim to continue and examine the studies using the chemically synthesized ZnO-NPs in the field on the same cultivar under different conditions, considering the complexity and importance of the field and the commerciality. Furthermore, we also are currently analyzing the transcriptome out of the leaves to get an extensive look at the subcellular alterations that may occurred to the plant leaves themselves. The RNA extraction and sequencing are ongoing, we also plan to validate the outcomes using the rtPCR quantitively and perform a technique for ROS imaging and localizing the zinc element in the tissues. It was an expanded study, and many analyses have been conducted through the authors, of course, it would be very fruitful to assess some other parameters. Since the study focused on determining the biochemical markers and the growth attributes in the presence or absence of salt/ZnO-NPs, we did not perform some useful analyses that obviously would improve the promising outcomes of the study. But, the work is ongoing. Thanks so much for your suggestion.
Minor comments
- For the macro and micro-elements, why not check the N and P contents?
Thanks a lot for your comment. We appreciate your concern. We planned two phases for our study. We planned to deal with leaves in the first phase. Determining what concerned us about Biochemistry through some factors and stress markers, some micro and macro elements, and also some growth and morphological attributes. We tried to assess the most important macro elements (from our point of view); Na, K, Mg. They were critical to be determined as Na was the main stress element in the study, K is the competitor of Na, and Mg is so important for chlorophyll. We also tried to estimate the most important microelements (from our point of view) like; Zn, Co, and Mn, as they are crucial in the study. Firstly, Zn determination was critical to investigate the success of leaf absorption to the chemically synthesized ZnO-NPs, and it is a really important element in biosynthesizing the amino acid tryptophan which plays a vital role in synthesizing the indole-based auxins, and also as a co-factor for the enzyme superoxide dismutase. Copper is also important as co-factor for SOD with the Zn element. Mn was also determined as a major contributor to various biological systems including photosynthesis, respiration, and nitrogen assimilation.
Secondly, for the other macro elements like Ca, or N and P, and even the microelements like Fe, we planned from the beginning to estimate them in the fruits, we may also re-determine the same elements that were determined in the leaves besides those desired ones in the fruits. We still have the intact fruits, we also have the productivity data of them and plan to go on another study regarding their effect as antimicrobials, antioxidants, and anticancer agents. So, we did not estimate the above-mentioned elements as we plan to estimate them with some others in the tomato fruits.
- The first letter of any keyword should be capital
Thanks for the notice. All the keywords were capitalized.
- Figures 11-16 show, show should be in the past tense showed, please check the others.
Thanks a lot. It was corrected. Noticing that; Figures 11-16 were transferred to the appendix. But, all the typos, grammar, and English language were revised again carefully.

Round 2
Reviewer 2 Report
Comments and Suggestions for Authors
I am glad that I helped the authors improve the text of the manuscript. I recommend it for publication.
Reviewer 3 Report
Comments and Suggestions for Authors
Dear Authors
I carefully reviewed the revised manuscript and the authors' responses to comments. Most of the recommendations were taken into account by the authors in the process of finalizing the manuscript. The manuscript has become more compact and easier to read. I think that it can be recommended for printing.
Kind regards
Reviewer 4 Report
Comments and Suggestions for Authors
Since the authors addressed all my questions, thus, I have no further comments.